# Review article: Design and Evaluation of Weather Index Insurance for Multi-Hazard Resilience and Food Insecurity

Marcos Roberto Benso[1], Gabriela Chiquito Gesualdo[1], Roberto Fray Silva[2,5,6], Greicelene Jesus Silva[1], Luis Miguel Rápalo[1], Fabrício Alonso Richmond Navarro[1], Patrícia Angelica Alves Marques[3,5], José Antônio Marengo[4,7], and Eduardo Mario Mendiondo[1,5,6]

[1]University of São Paulo, São Carlos School of Engineering, São Carlos, Brazil
[2]University of São Paulo, Institute of Advanced Studies, São Paulo, Brazil
[3]University of São Paulo, Luiz de Queiroz College of Agriculture, Brazil
[4]Brazilian National Center for Monitoring and Early Warning of Natural Disasters (CEMADEN), São José dos Campos, São Paulo, Brazil
[5]Center for Artificial Intelligence - AGRIBIO (C4AI-AGRIBIO), Brazil
[6]National Institute of Science and Technology for the Fight Against Hunger, Brazil
[7]National Institute of Science and Technology for Climate Change Phase 2 (INCT-II), Brazil

**Correspondence:** Marcos R Benso (marcosbenso@gmail.com)

**Abstract.** Ensuring food security against climate risks has been a growing challenge recently. Weather index insurance has been pointed out as a tool for increasing the financial resilience of food production. However, the multi-hazard insurance design needs to be better understood. This paper aims to review weather index insurance design for food security resilience, including the methodology for calculating natural hazards' indices, vulnerability assessment, and risk pricing. We searched for relevant research papers in the Scopus database using the Preferred Reporting Items for Systematic reviews and Meta-Analyses (PRISMA) protocol. Initially, 364 peer-reviewed papers from 01/01/2010 to 19/02/2022 were screened for bibliometric analysis. Then, the 26 most relevant papers from the last five years were systematically analyzed. Our results demonstrate that despite a significant research effort on index insurance, most papers focused on food production. However, research considering other aspects of food security, such as transportation, storage, and distribution, is lacking. Most research focuses on droughts. Other hazards, such as extreme temperature variation, excessive rainfall, and wildfires, were poorly covered. Most studies considered only single-hazard risk, and the multi-hazard risk studies assumed independence between hazards, neglecting the synergy hypothesis between hazards. Lastly, we proposed a conceptual framework that illustrates design paths for a generalized weather index insurance design and evaluation. Solutions on how to address multi-hazard problems are considered. An illustrative example demonstrates the importance of testing the multi-hazard risks hypothesis for weather-based index insurance design for soybean production in Brazil.

## 1 Introduction

The increased frequency and magnitude of extreme weather and climate events have been evidenced in many regions of the globe, being widely attributed to climate change (IPCC, 2022). In recent years, extreme weather events have caused significant

losses and damages in many climate-sensitive sectors affecting urban and rural areas. Insurance is essential to provide economic sustainability to vulnerable sectors and improve recovery from catastrophic climate events.

Insurance has been pointed out as a tool for safeguarding populations and properties from climate change (UNEP, 2012). Nevertheless, Kraehnert et al. (2021) argue that insurance itself is not an adaptation measure and depends on several characteristics and factors. Some relevant factors are living standards, economic well-being, the availability of safety nets for poor people, characteristics of the sector, and the type of risks to which sectors are exposed (FAO, 2014).

Swiss Re (2021a) predicted non-life insurance premiums to rise 10% above the pre-pandemic state and acknowledges that climate change might have an even more significant impact on the insurance industry. They propose that increasing underwriting policies against climate-related disasters is vital to tackle this problem.

However, the challenge might be more significant in developing countries with lower insurance coverage. On the one hand, the premiums per capita in the US and Canada were 7,270 USD in 2020 much higher than the world average of 809 USD per capita and the Eurozone average of 2,723 USD. Conversely, Latin America, the Caribbean, emerging Europe, and Asia presented premiums of 203, 159, and 215 USD per capita, respectively. Africa and the Emerging Middle East presented much lower numbers, of 45 and 93 USD per capita, respectively (Swiss Re, 2021b).

Index-based insurance policy is a solution to improve insurance coverage, especially in low-income areas (Raucci et al., 2019). The term index insurance started being used for crop yield insurance policies based on area-yield indices as firstly described by Halcrow (1949) and then further revisited by Miranda (1991). The area-yield insurance model was adopted in the US in the early 90s, dividing agricultural areas in the crop domain into Group Risk Plans (GRP). Indemnities were triggered when forecasted crop yields would fall under a certain threshold within each GRP (Skees, 2008).

Area-yield contracts depend on data availability and technical capacity to evaluate and monitor the group risk units, which can be costly and impractical in many poor and developing countries. To overcome this challenge, researchers proposed contracts based on weather indices (Müller and Grandi, 2000).

In the financial and actuarial literature, weather derivatives have been used to associate the financial frustration of a business with a weather index (Müller and Grandi, 2000). Contracts based on weather indices have helped policyholders to hedge against adverse conditions in the clothing business (Štulec et al., 2019a), hydropower plants (Foster et al., 2015), and solar energy systems (Boyle et al., 2021). Crop yield contracts based on rainfall have been used due to their simplicity and data availability (Yoshida et al., 2019). The method uses rainfall from weather stations nearby farms to predict losses, and the threshold is usually defined according to an index in the growing season.

This type of contract almost eliminates the need for on-site verification of losses, reducing administrative costs and improving the transparency of insurance products (Shirsath et al., 2019). Insurance companies also benefit from reducing moral hazards since crop losses are estimated from indices provided by third-party agencies (Ghosh et al., 2021). Moreover, due to reduced costs, contracts based on weather indices have been used for microinsurance contracts in poor rural areas to improve protection against adverse climate conditions and prevent smallholder farmers from falling into poverty traps (Skees, 2008). Despite its advantages, index insurance has a particular side effect called basis risk, which is a mismatch between actual losses and predicted losses (Ghosh et al., 2021).

As expected from the relevance of agriculture in the insurance industry, most literature reviews focus on understanding index insurance and microinsurance for agriculture (Leblois et al., 2014; Sarris, 2013). Zara (2010) proposed a systematic review of the role of weather derivatives in the wine industry. Akter (2012) focused on reviewing problems of microinsurance in Bangladesh, looking for evidence for insurance demand, how to approach the market, and design challenges to improve the safety of the vulnerable population, especially for smallholder farmers.

Several studies have been reported on single-hazard risk insurance design. Considering only one hazard does not include the expression of risk due to interactions among different hazards (Gill and Malamud, 2014; Hillier et al., 2020). The insurance risk assessment and climate change impacts have been recently reviewed by Lyubchich et al. (2019). The authors review several adverse events such as floods, hail, and excessive wind, but the interaction effect between hazards could be further explored.

Sekhri et al. (2020) proposed a framework for multi-hazard risk management. However, it was too specific for mountainous regions and a broader risk management strategy. Komendantova et al. (2014) introduced a framework for participatory risk governance, allowing for feedback from stakeholders. Abdi et al. (2022) conducted an extensive review of the possible index insurance applications for agriculture. The authors summarized indices and methods for designing index insurance with possible applications for multi-hazard risks. However, multi-hazard implementation has not been nearly as thoroughly investigated as single-hazard problems.

Considering this initial analysis, this paper thoroughly analyzes the literature, further describes the identified gaps, and proposes a framework for addressing multi-hazard index-based insurance design for agricultural purposes. The systematic review was designed to answer the following questions, considering the context of index insurance: 1) What indices are used to assess and monitor extreme weather events?; 2) What functions and methods are used to assess the vulnerability of food production to extreme weather events?; and 3) How to determine risk premiums?

The paper is organized into the following sections: section 2 presents the methodology used to conduct the systematic literature review; section 3 reports the main findings of the literature review, discusses the most relevant papers, presents the proposed framework for insurance design, and illustrates its use with an example for soybean production in Brazil; and section 4 concludes the paper, pointing out limitations and recommended future works.

## 2 Methodology

A systematic review was conducted to better identify the state-of-the-art in designing and implementing multi-hazard index-based insurance in agricultural environments and to identify the main gaps in the current techniques and models. The Preferred Reporting Items for Systematic reviews and Meta-Analyses (PRISMA) protocol (Liberati et al., 2009) was applied, and the Scopus database was used for data collection.

This database was chosen due to its comprehensive coverage of relevant events and scientific journals related to climate change, agriculture, insurance design, and multi-hazard frameworks and techniques, among other relevant topics. It encompasses a wide range of subjects in technology, science, social sciences, medicine, humanities, and arts (Scopus, 2022).

We performed the literature review following the PRISMA protocol (Liberati et al., 2009). First, a bibliometric analysis was performed on the selected papers from 01/01/2010 to 19/02/2022, using the Bibliometrix R package (Aria and Cuccurullo, 2017). Then, a critical analysis of the most cited papers of the last five years (2018 to 2022) was performed to identify fundamental research topics, themes, keywords, and guidelines for index insurance design and evaluation and to identify the main gaps in the literature.

The systematic review process was divided into four steps (Figure 1). The first consisted of defining the search strings based on the three research questions described in section 1. Our search string was composed of keywords in the English language extracted from an in-depth analysis of relevant literature reviews and papers on the topic. It was then used to search terms in the documents' title, abstract, and keywords in the Scopus database. The following criteria were considered:

- English keywords: multi-risk weather index insurance.

- English synonyms: multi-risk, risk, weather, climate, index, parametric, insurance, microinsurance, derivative.

- Search string: TITLE-ABS-KEY ( (risk (multi AND risk) OR portfolio) OR ( index OR parametric ) AND ( insurance OR microinsurance OR derivative ) AND ( weather OR climate ) ).

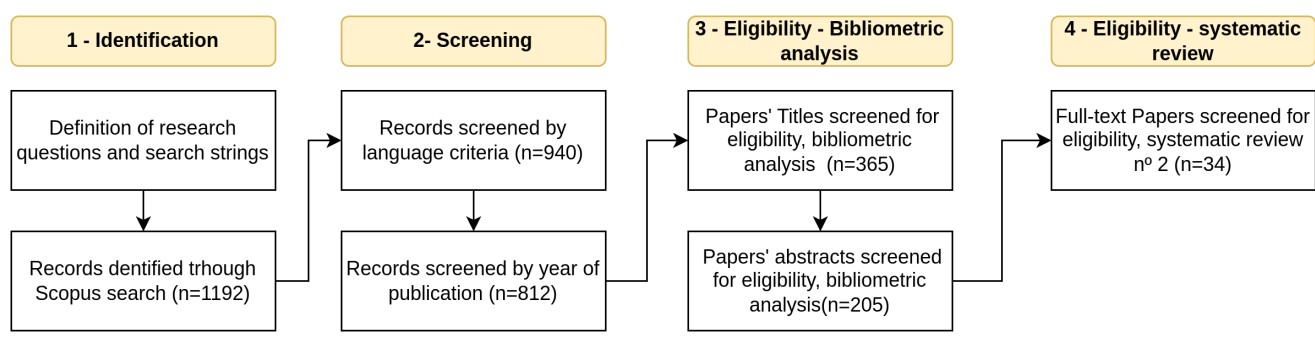

**Figure 1.** Methodological steps of PRISMA statement

The second step was the screening process. First, we selected only scientific papers published in peer-reviewed journals in English, Portuguese, or Spanish. Review papers, books, book chapters, and conference proceedings were excluded from the analysis, following the methodology used in other systematic reviews in the literature. This step resulted in 1192 documents.

In the third step, an analysis of the documents' titles and abstracts was conducted to filter only works that designed or implemented a complete application of an index insurance or weather derivative. Many studies on the evaluation of index insurance demand and traditional insurance models were excluded. This step resulted in 365 documents. Then, several tools used for bibliometric analysis were applied to this dataset.

The fourth step was performing a critical review of the 26 most cited papers published in the last five years of the dataset (2018 to 2022). This evaluation excluded papers that did not provide information on index insurance design. This review was

divided into: (i) hazard identification; (ii) vulnerability analysis; and (iii) financial method and risk pricing analyses. These three modules were adapted from the frameworks developed by Guzmán et al. (2020), Mohor and Mendiondo (2017), and Righetto et al. (2007), which encompass the main aspects of weather-based insurance design.

Before analyzing the full papers, it is critical to specify the main concepts and definitions. Although there are many definitions for concepts such as hazard, multi-hazard, resilience, and food security, we chose to adopt the most broadly accepted ones. These were:

– Hazard: "A dangerous phenomenon, substance, human activity or condition that may cause loss of life, injury or other health impacts, property damage, loss of livelihoods and services, social and economic disruption, or environmental damage" (UNDRR, 2014). This paper refers explicitly to hazards derived from extreme weather and climate events.

– Multi-hazard: "[...] all possible and relevant hazards and the valid comparison of their contributions to hazard potential, including the contribution to hazard potential from hazard interactions and spatial/temporal coincidence of hazards, while also taking into account the dynamic nature of vulnerability to multiple stresses" (Gill and Malamud, 2014).

– Vulnerability: "The conditions determined by physical, social, economic and environmental factors or processes, which increase the susceptibility of a community to the impact of hazards" as defined by the Hyogo Framework for Action (UNDRR, 2014). For this paper, the concept of vulnerability was focused on the physical damages and losses derived from the realization of an extreme weather event. We are utilizing, therefore, a classical approach to quantify the vulnerability of risk-averse individuals, which considers that the greater the losses, the more the vulnerability. Even though this traditional definition has been questioned as a reducer of solely the economic sphere of an issue that permeates social, political, and environmental dimensions, this is ultimately a practical approach of widespread use (Machado et al., 2005).

– Resilience: "The ability of a system, community or society exposed to hazards to resist, absorb, accommodate to and recover from the effects of a hazard in a timely and efficient manner, including through the preservation and restoration of its essential basic structures and functions" UNDRR (2009). In the context of this paper, and as described primarily by Mohor and Mendiondo (2017) and Guzmán et al. (2020), in the resilience module of an index insurance schema, the risk premium is an indicator of the resilience of a sector for coping with weather and climate extreme events.

– Food security: "exists when all people, at all times, have physical, social and economic access to sufficient, safe and nutritious food which meets their dietary needs and food preferences for an active and healthy life. Household food security is the application of this concept to the family level, with individuals within households as the focus of concern." (FAO, 2003). In summary, food security is rooted in the pillars of availability, access, and utilization (Barrett, 2010). This broadens the concept of food security to encompass different supply chain links, such as food production, transportation, storage, and distribution.

# 3  Results and Discussion

This section describes the main results of this work. It also discusses important aspects related to applying these results in different scenarios and contexts. It is divided into the following subsections: 3.1 contains the main results of the bibliometric analysis; 3.2 presents an in-depth literature review of the most relevant papers identified, exploring the hazard assessment, vulnerability analysis, and financial methods and risk pricing modules; and 3.3 presents the proposed conceptual framework, encompassing both its description and an example to illustrate its main aspects and contributions to the field of multi-hazard weather-based insurance design.

## 3.1  Bibliometric analysis

First, it is vital to observe that around 50% of the works analyzed were published since 2018, denoting the increased interest in the topic. The average number of citations per year per paper demonstrates an increasing impact of weather index insurance in the literature (Figure 2a). However, the global distribution is concentrated in Europe, USA/Canada, and Asia. The role of Latin America/Caribbean, Australia/New Zealand/Oceania, and Africa are much lower, representing less than 10% of the published papers each.

Additionally, international collaboration is a critical factor for high-impact scientific studies. Two important countries to analyze in this aspect are Russia and China. In Russia, more than 90% of highly cited papers were written in an international setting (Pislyakov and Shukshina, 2014). Similarly, in China, 47% of highly cited papers were written in an international collaborative form.

These countries' international cooperative background, in general, opens the way to more innovative research in the field. These publications illustrate how collaborations with international scientists from centers of excellence enhance the study's dissemination.

The scientific collaboration map (Figure 2b) shows strong collaboration networks between the United States, European countries, China, and India. European countries such as Germany, Switzerland, and the Netherlands have played a dominant position in integration and have promoted collaboration with Kenia, Ethiopia, Nigeria, and South Africa. Canada has collaborated with China, Indonesia, the United States, and European countries. From this analysis, we conclude that the United States, China, and Germany play dominant positions in scientific collaboration and are the most influential countries.

A keyword analysis revealed that agricultural and crop insurance are well-developed subjects with a substantial impact on index insurance. In addition, drought is the most studied hazard, explained by the impacts of droughts on agriculture. It is important to note that since index insurance was designed to be used in agriculture (Miranda, 1991; Skees, 2008) and, as the concept has gained attention, a broader range of applications might be proven feasible.

On the one hand, Latin American countries such as Brazil, Argentina, and Mexico are vital in global food production (Baldos et al., 2020). On the other, by conducting a bibliometric analysis of relevant studies from 2010 to 2022, we discovered a low academic engagement between them and the rest of the world. This is incoherent, as these countries suffer the most from extreme events losses due to their solid economic link with climate-dependent primary activities. These findings emphasize

the importance of developing index insurance in tropical countries, notably Latin America, to adapt better to climate change. Furthermore, climate change and basis risk are critical in building index insurance. However, these themes need to be developed more in the literature analyzed.

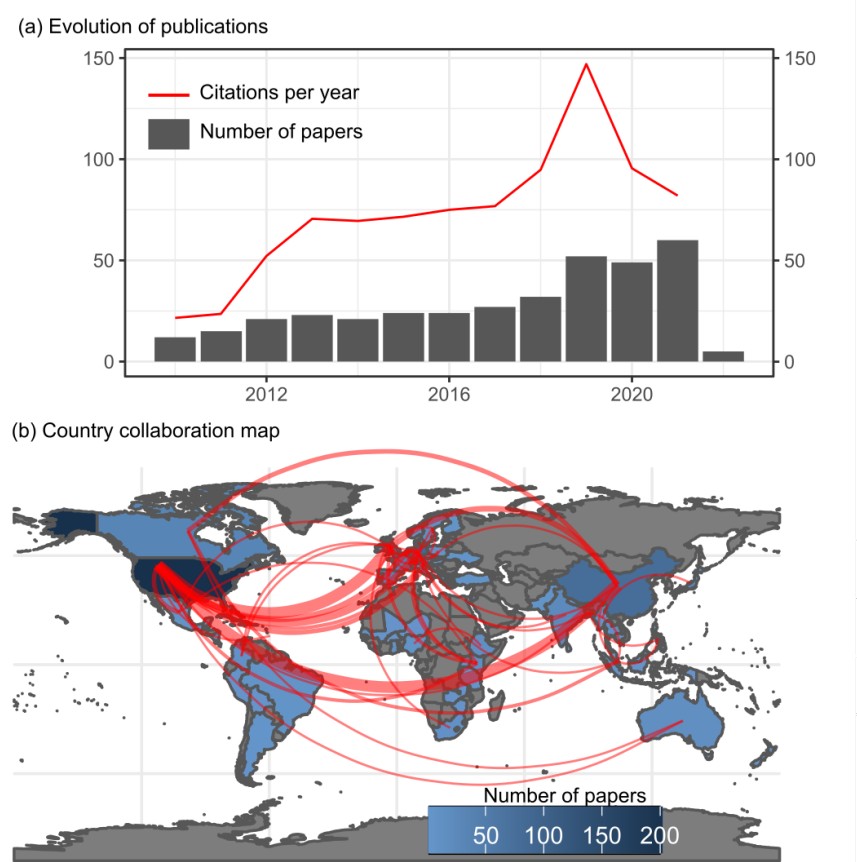

**Figure 2.** Weather index insurance studies. (a) Temporal distribution from papers collected from 01/01/2010 to 19/02/2022. (b) Thematic map representing the global collaboration network, where the countries in blue represent the number of studies produced by scientists. The darker the color, the more affiliations. The world vector map data was provided by https://www.naturalearthdata.com/ under public domain.

Table 1 presents a strategic diagram to analyze clusters of keywords (referred to as themes) according to their centrality and density values. According to Cobo et al. (2011) density is a measure of the development of the theme, and centrality is the importance of the theme in the development of the whole field of research we analyzed. According to their densities and centralities, the themes are divided into four classes: basic, motor, niche, and emerging and declining themes. These will be analyzed in the following paragraphs.

Basic themes represent relevant keyword clusters for all the documents analyzed. These include the following clusters: "index-based insurance", "climate change", "index insurance", and "basis risk". Climate change has been a significant concern for decision-makers, especially risk management. Climate change might lead some regions toward higher risk profiles, increas-

ing their vulnerability and the expected losses. Therefore, this theme represents an opportunity to develop index insurance for agriculture. Basis risk is a primary topic that requires more development. Even thoughtit is a well-known bottleneck in the field, and our analysis suggests room for improvement. More attention to this topic must be paid to in future studies.

Motor themes encompass keywords that are relevant to the entire insurance theme and that are well-developed. Since they present strong centrality and high density, the clusters "insurance", "agriculture" and "risk management" and the cluster "crop insurance" are conceptually related to almost all papers gathered in the bibliometric analysis. This result confirms that agricultural and crop insurance are the most explored themes in the index insurance field.

    Emerging and declining themes are related to themes that represent the combination of low levels of development and are

marginal to the entire field of research. This quadrant includes "weather index insurance", which is a critical issue in terms of impact since extreme weather events trigger significant disasters worldwide. Given the need to manage weather extremes and their importance to a broader geophysical community, weather index insurance is an emerging topic that will gain more attention in the following years.

    Finally, niche themes encompassed the term "weather derivatives". It is considered a niche theme because it is well-developed

but has a marginal impact in the field. The themes have fundamental distinctions and similarities. Derivatives are traded Over the Counter (OTC) or on Chicago Mercantile Exchange (CME). Index insurance is a product offered by insurance and reinsurance companies. They theoretically have a similar principle: a risk-averse individual pays a premium for a risk-bearing individual.

**Table 1.** Thematic mapping of the documents based on the conceptual structure of the author's keywords divided into seven clusters with word frequency higher than 40 words according to centrality (the relevance of the theme in the development of the field) and density (the development of the field)

| Theme | Cluster | Density | Centrality |
|---|---|---|---|
| Motor | Insurance | 9 | 7 |
| | Crop insurance | 7 | 5 |
| Niche | Weather derivative | 2 | 8 |
| Emerging and Declining | Weather index insurance | 3 | 4 |
| | Index-based insurance | 5 | 3 |
| Basic | Basis risk | 8 | 1 |
| | Climate change | 6 | 2 |

### 3.2   Systematic literature review

The food security concept is rooted in the pillars of availability, access, and utilization (Barrett, 2010). Food production can affect availability, while access to renewable or sustainable energy can facilitate proper food transportation and storage. Not all research in the systematic review could be thoroughly examined due to a lack of information on their applications. Thus, for the

26 papers with complete information, we conducted an overview of the application and most relevant characteristics of index insurance for food security in three main categories: (i) agricultural; (ii) hydrological; and (iii) sustainable energy insurance. Table 2 presents the results of this analysis.

We observed that most studies evaluated insurance at different spatiotemporal aggregations, such as crop insurance, which was analyzed at the farm level by governmental agencies, insurance companies, or surveys. In many countries, for example, agricultural data is aggregated at a regional scale, i.e., municipality, department, state, and country, without standardization. The size of the properties varied greatly, from 5 to 400 hectares and the total coverage was up to 1.6 million hectares.

Forestry insurance covers larger areas and uses remote sensing data to assess risk. Therefore, spatial discretization is performed at a pixel level. The catchment level was the spatial unit for hydrological insurance, and the coverage included all hydrological processes that occurred upstream of the reservoir. For the sustainable energy insurance - wind and solar power insurance - a unique point, representing the location of the windmills and solar panels, was evaluated

The temporal scale in which the insurance was purchased varied from seasonal to annual. Crop insurance is typically contracted before the sowing period and reaches maturity at the end of the crop cycle. Sectors that are continuously exposed to natural hazards are operated on an annual basis.

The insurance premiums were represented using different units. However, most of the works focused on premiums per unit of area and unit of cost. The crop insurance premium varied from $6.18 to $55.26 USD per hectare. This value was affected mainly by the cost of production and farmers' degree of risk aversion. A value of $187.29 USD per ton of crop and 3 to 7% of production costs was also found.

In contrast, the hydrological insurance for water supply represents values of $10.48, and the irrigation insurance ranges from $212.83 to $333.07 USD per hectare. The prices for irrigation were inconsistent with crop insurance. This might be related to irrigation costs. Sustainable insurance presented premium rates ranging from 0.35 to 0.50 of production costs or a percentage of $0.033144 per kWh. A detailed analysis of hazard identification, vulnerability analysis, and financial methods of the reviewed paper is presented in sequence.

### 3.2.1   Hazard assessment

The index insurance literature contains a broader range of analyzed hazards and indices (Table 2). Drought and hydrological drought are the most frequent hazard (77%), followed by excessive rainfall and flood (27%), temperature variation: heat and cold waves (4%), wildfire (4%), low wind speed (4%), and lack of solar radiation (4%). Furthermore, 27% of the studies have a multi-hazard interaction, with drought and excessive rainfall being the most common, followed by wildfire and excessive rainfall, and hydrological drought and floods.

Most studies focused on drought, which is consistent with the findings of Abdi et al. (2022), who identified drought as the leading risk when studying index insurance for crop productions. This is due to the fact that drought is the most damaging threat in the agricultural sector, and the sector was the driving subject of the studies evaluated. Our finding is consistent with the fact that drought-induced yield losses occurred in three-fourths of the global harvested regions between 1983 and 2009 (Kim et al., 2019).

**Table 2.** Main categories for index-based insurance and specific application. Indices: Cumulative Precipitation Index (CPI); Water Storage (WS); Water Deficit (WD); Normalized Difference Vegetation Index (NDVI); Soil Moisture Index (SMI); Standardized precipitation evapotranspiration index (SPEI); Standardized Precipitation Index (SPI); El Niño Southern Oscillation (ENSO); Evaporative Stress Index (ESI); Ped Drought Index (PDI); Ribéreau-Gayon and Peynaud hydrothermal scale (RGP); R2mm; Berman and Levadoux (BBL); High Temperature (HT); Low temperature (LT); Ribéreau-Gayon and peynaud hydrothermal scale (RGP); Visible Infrared Imaging Radiometer Suite (VIIRS); Solar Radiation (SR); Wind Speed (WSpeed)

| Category | Hazard | Index | Threshold | Author |
|---|---|---|---|---|
| Agricultural | Drought | CPI | Expected yield | Turvey et al. (2019) |
| | | | Expected yield | Roznik et al. (2019) |
| | | | 30th percentile | Kath et al. (2019) |
| | | | 50-85th percentile | Awondo (2019) |
| | | | Expected yield | Ricome et al. (2017) |
| | | CPI, NDVI | Customized trigger | Eze et al. (2020) |
| | | CPI, SPI, SPEI, SMI, ESI | Expected yield | Bucheli et al. (2021) |
| | | ENSO | Expected yield | Mortensen and Block (2018) |
| | | PDI, CPI | Expected yield | Bokusheva (2018) |
| | | SMI | Expected yield | Vroege et al. |
| | | SPI, SPEI | Expected yield | Hohl et al. (2020) |
| | | WD | Expected yield | Gómez-Limón (2020) |
| | Drought and excessive rainfall | BBL, RGP | 50th-90th percentile | Martínez Salgueiro (2019) |
| | | CPI | 50th percentile | Martínez-Salgueiro and Tarrazon-Rodon (2020) |
| | | CPI, R2mm | Customized trigger | Shirsath et al. (2019) |
| | Drought and excessive rainfall | DOWKI | 10th and 90th percentile | Kapsambelis et al. (2019) |
| | Excessive rainfall | CPI | 94th percentile | Furuya et al. (2021) |
| | | | 70th - 95th percentile | Kath et al. (2018) |
| | Temperature variation (high and low) | HT, LT | 5-year moving average yield | Guo et al. (2019) |
| | Wildfires and excessive rainfall | WSpeed and VIIRS | Customized trigger | Sacchelli et al. (2018) |
| Hydrological | Drought | WD | Q710 | Mohor and Mendiondo (2017) |
| | | | 70th-90th percentile | Denaro et al. (2018) |
| | Hydrological drought | WS | Expected yield | Guerrero-Baena and Gómez-Limón (2019) |
| | Hydrological droughts and floods | WD, PF | Expected yield | Denaro et al. (2020) |
| Sustainable Energy | Lack of solar ratiation | SR | Customized trigger | Boyle et al. (2021) |
| | Low wind speed | WSpeed | Customized trigger | Rodríguez et al. (2021) |

Index insurance is a promising methodology for designing insurance models because it avoids the high administration costs, adverse selection, and moral hazard issues associated with traditional indemnity-based insurance. The behavior of an index can characterize the variability of a hazard. However, its performance in covering losses is highly dependent on the index chosen.

The index choice is a critical phase in index insurance modeling since a mismatch between the index and the actual loss might increase basis risk. Another significant element influencing performance is geographic basis risk, which occurs when the insurance index is based on a location other than the insured location. This difficulty emerges, for example, in agricultural insurance when the index is derived from a meteorological station placed in a location that does not adequately represent the insured region.

Because it is challenging to give specific index insurance contracts to small regions, geographical basis risk is generally unavoidable (Odening and Shen, 2014). Another option for mitigating this risk is to utilize decorrelation functions and an insurance portfolio composed of contracts for various regions (Norton et al., 2013). Figueiredo et al. (2018) proposes a probabilistic framework for tackling uncertainties in trigger selection and damage modeling, therefore improving basis risk quantification, evaluation and communication.

Approximately 50% of the studies analyzed used a rainfall-based drought indicator (i.e., CPI, SPI, CDD, and R2mm) to indicate drought and excessive rainfall. This straightforward strategy indicates drought conditions and only needs precipitation data. Besides, these indices have the advantage of representing both water deficit and water excess.

Rainfall indices in the agricultural sector might considerably represent low-yield occurrences, in both deficit and excess forms, which would correlate well with low-yields (Abdi et al., 2022). Indices such as the Standardized Precipitation Index (SPI) have the advantage of being determinable even when there are gaps in the data. However, the SPI's application is limited when issues related to the water balance must be considered in the problem analysis.

As an alternative, the Standardized Precipitation Evapotranspiration Index (SPEI), while capable of reflecting the combined effect of rainfall and temperature variations on drought, requires data for radiation, temperature, and relative humidity. These can be challenging to obtain in developing countries with low density of weather stations.

In addition to data availability, the index's simplicity of computation is a significant factor to consider while choosing it. As a result of their ease of calculation and data input, indices such as CPI, Consecutive dry days (CDD), Water Storage (WS), and Water Deficit (WD) are popular among the studies examined. While data for CPI, CDD, WS, and WD calculations are easily acquired, the spatial distribution of risk has yet to be fully known, even in places with long-established weather stations. This poses a difficulty in the index insurance market: strategically exploiting information from existing stations at geographic locations where the precise weather observations are unknown (Norton et al., 2013).

The Standardized Precipitation Index (SPI) and Standardized Precipitation Index (SPEI) are more complicated indexes that add weather statistics to the analysis. Aside from the complexity of its calculation, these indices require a database of at least 30 years. Indices derived from form modeling techniques and remote sensing are much more challenging to acquire and determine.

Besides that, complex indices, such as the Soil Moisture Index (SMI) and the Normalized Difference Vegetation Index (NDVI), as well as the usage of El Nino Southern Oscillation (ENSO), have grown more frequent as modeling techniques

and remote sensing have advanced. This popularization is already seen in the current review, as they were used in 11% of the studies investigated.

A disadvantage of using indices derived from remote sensing is that these datasets have shorter time series. At least 20 years of data is required to understand historical patterns and to provide more confidence for historical burn rate analysis pricing (Norton et al., 2013).

While some indices are more popular than others, it is crucial to realize that no index can be used for all contexts and situations. The available data, degree of drought monitoring, and time resources available for its determination all influence the index representation of the hazard, as long as the uncertainties (basis risks) are under control.

Other hazards encompass temperature variation, wildfire, low wind speed, and lack of solar radiation, accounting for 16% of the studies analyzed. Thermal hazard is a growing concern for human health, agriculture production, forestry, and the environment. Similar to rainfall, temperature extremes can also explain satisfactory yield losses (Abdi et al., 2022).

Given the topic's relevance, we expected more work focusing on this subject. However, we have found only one study on temperature variation insurance. High-Temperature Index (HTI) and Low-Temperature Index (LTI) were proposed by Guo et al. (2019) that focused on computing the number of days the temperature was higher or lower than a certain threshold, representing the rice yield reduction. Regardless of the hazard, we had one study per index. This reflects the diversity of these issues and a need for in-depth research on the specific hazards - temperature variation, fire, wind, and solar radiation.

The hazards are treated as independent occurrences in the multi-hazard interaction. However, not every study examined one index per hazard. More than half of the multi-hazard studies considered drought and excessive rainfall, and they used the same index to reflect both hazards (I.e., CPI, DOWKI, and R2mm). According to Kapsambelis et al. (2019), simple climatic water balance based on precipitation and evapotranspiration data can simulate both drought and excess rainfall globally.

The authors state that the DOWKI index was able to fit extreme yield anomalies. One example of employing different indexes to reflect different hazards was found on wildfires and excessive rainfall on forestry in Italy (Sacchelli et al., 2018). The authors described forest fires using the Visible Infrared Imaging Radiometer Suite (VIIRS) and explored the effects of strong winds through wind speed (Wspeed).

The advantage of using only one index to represent two hazards is the ease of calculating and implementing the insurance policy. However, not all hazards can be represented by a single index, such as wildfires and excessive rainfall. It is known that multi-hazards and compound events are increasingly intense and significant. The finding of only a few studies connected to the theme reflects a substantial gap in the literature.

### 3.2.2 Vulnerability analysis

In finance and management, insurance is a product that intends to totally or partially reduce or eliminate the loss caused due to different risks (Ejiyi et al., 2022). In order to lower the basis risk and boost the acceptance of insurance products, the prediction models of the link with the index and the damages must be effectively chosen. The vulnerability analysis is focused on modeling the physical damages and losses from an extreme weather event. Thus, we presented a summary of the Expected

**Table 3.** Summary of Expected Loss Amount (ELA) and Expected Annual Damage (EAD) models

| Type of impact | Type of loss model | Authors |
|---|---|---|
| | Cluster analysis | Eze et al. (2020) |
| Expected loss amount (ELA) | Linear regression | Bokusheva (2018) |
| | | Bucheli et al. (2021) |
| | | Furuya et al. (2021) |
| | | Gómez-Limón (2020) |
| | | Guerrero-Baena and Gómez-Limón (2019) |
| | | Guo et al. (2019) |
| | | Hohl et al. (2020) |
| | | Kath et al. (2019) |
| | | Mortensen and Block (2018) |
| | | Denaro et al. (2020) |
| | GALM | Awondo (2019) |
| | | Kath et al. (2018) |
| | Stepwise regression | Shirsath et al. (2019) |
| | Copula | Bokusheva (2018) |
| | | Kapsambelis et al. (2019) |
| | | Martínez Salgueiro (2019) |
| Expected annual damage (EAD) | Empirical curve | Mohor and Mendiondo (2017) |
| | | Sacchelli et al. (2018) |

Loss Amount (ELA) and Expected Annual Damage (EAD) models applied in the reviewed papers, available in Table 3. The deterministic models were applied for income reduction impacts, especially crop insurance.

Most applied models were related to a unique explanatory variable or index. Due to their simple application and understanding, these models are expected to be the most common, primarily linear regression models. However, these models present the disadvantage of contemplating only one hazard at a time. In contrast, the Generalized Additive Linear Models (GALM) and stepwise regression added the possibility of evaluating more than one index. These can then be used in a multi-hazard approach.

A multi-hazard approach requires understanding the frequency and magnitude of multiple hazards and the possibility of them occurring simultaneously. The multi-hazard risk index insurance papers analyzed presented combinations of drought and excessive rainfall for crop insurance (Kapsambelis et al., 2019; Shirsath et al., 2019), fire and storms for forestry insurance (Sacchelli et al., 2018), temperature variation and excessive rainfall for crop insurance (Martínez Salgueiro, 2019), and high and low temperatures for crop insurance (Guo et al., 2019). The assumption of independence was considered prior knowledge by Martínez Salgueiro (2019) and Guo et al. (2019). However, the authors did not provide mathematical proof of this choice. Instead, they prioritized hazards according to their frequency and magnitude using pre-existent risk maps.

Another possibility to incorporate hazard interactions is through Copulas. This has been incorporated in the loss modeling by Kapsambelis et al. (2019) and Martínez Salgueiro (2019). The copula theory (Nelsen, 2006) is widely used for multi-hazard analysis since it derives joint probability distributions from marginal distributions. Briefly, the marginal distributions are not required to follow the same probability distribution model, giving flexibility and robustness to analyze the interaction of more than two marginal distributions.

Additionally, losses and damages databases are not necessarily Gaussian normally distributed in copulas models, which is common in crop production. This way, a significant amount of skewed information can be well embodied, depicting better hydrological and meteorological extremes than linear regression models.

In addition, to embody loss and damages originating from multi-hazard events, another important aspect of vulnerability assessment is the representation of complex patterns in the loss and damage series modeling. In this regard, linear regressions are commonly applied in loss forecasting due to their simplicity.

Generalized additive linear models (GALMs) add a link function to express a linear relationship between more than one variable (Blier-Wong et al., 2020), making it possible to express both multi-hazard risk phenomena and simple nonlinear effects. While simple and easily explainable, such models may be ineffective in learning complex patterns in the data, which are common in food production. Machine learning (ML) can enable the optimal formulation of insurance policies when applied to the insurance field. These models help to capture high-dimensional, nonlinear, and complex interactions between indices and losses (Blier-Wong et al., 2020).

ML techniques are still emerging in loss models, and here we have reviewed only a very recent paper (from 2020) that used cluster analysis. The paper provides evidence that ML techniques can improve loss modeling from different sources and present different time and spatial scales (Eze et al., 2020). Blier-Wong et al. (2020) emphasized that ML applications in actuarial science are expanding rapidly and show great promise.

Frees et al. (2014) affirm that, with greater data availability and robustness in ML algorithms, more heterogeneity represented by insured individuals can be captured, representing their vulnerability accurately. With these models becoming more common in future studies, the multi-hazard assessment could be better incorporated. While promising, applying ML techniques to model loss and damages in the insurance sector may be bottle-necked, especially in developing countries, by the need for qualified personal and powerful GPUs.

When vulnerability studies or datasets for loss and damage quantification in specific sites are unavailable, the insurer can lay hands on empirical functions or crop modeling techniques. Monteleone et al. (2022) reviewed methods used to model the functional relationship between a given extreme event and crop losses. They also highlight the need for studying crop vulnerability to other less studied climate-related hazards, such as extreme temperatures. Regarding index insurance design, we found a paper that presented empirical functions based on the assumption of linearity between water deficit an losses. Mohor and Mendiondo (2017) used the water deficit volume to the $Q_{7,10}$ reference flow for predicting the impact of water shortage on water supply, irrigation, livestock, and ecological sectors. In crop insurance, if the historical yield losses database is non-nexistent or available at a high level of aggregation, crop modeling shows promise in estimating yield while utilizing different explainable variables.

### 3.2.3 Financial methods and risk pricing

The impact provided in the vulnerability analysis module can be translated as expected values of damage, income reduction, or business interruption by financial methods. The reviewed papers presented burning rate, probabilistic fit, and index modeling as the most prevalent risk pricing models. They commonly use the mean historical losses to estimate expected future losses for similar sectors (Sant, 1980).

The expected losses are called pure risk premiums and are the primary concern in index insurance papers. The historical losses are converted into payouts considering two critical variables (i) strike value K, and (ii) degree of coverage dc. The K is the index value that triggers payouts, proportional to risk aversion and the degree of coverage. The risk aversion is reflected in the degree of coverage, e.g., dc ranging from 0 to 1, being 0 with no protection and 1 with complete protection. These variables represent the behavior and aversion of policyholders towards a particular risk and will be the key to define the premium and indemnity values.

The loss expectation can be determined using the historical burn rate method (HBR), which is based on the observation of historical losses (Guerrero-Baena and Gómez-Limón, 2019; Hohl et al., 2020; Mortensen and Block, 2018; Shirsath et al., 2019). This method is widely applied in the insurance industry. However, it requires sufficient data in order to be accurate. For smaller datasets considering uncertainty, expected values can be evaluated by fitting loss data to a probability density function (Bokusheva, 2018; Bucheli et al., 2021; Kath et al., 2019; Eze et al., 2020; Kath et al., 2019; Martínez Salgueiro, 2019; Sacchelli et al., 2018; Vroege et al.). This procedure helps to improve pure risk premium rates by accounting for the probability of extreme events that have not been recorded. The probability distribution of loss data presents distortions in the tails, leading to underestimating pure risk premiums.

Moreover, insurance companies present nontraded assets that add costs to final premium rates. The transformation method proposed by Wang (2002), also referred to as Wang Transform, takes into consideration the impact of nontraded assets in premium rates, and the method was applied by Boyle et al. (2021) and Denaro et al. (2018).

Other approaches for defining contract payouts are based on a probabilistic fit. Bokusheva (2018) applied the Marginal Expected Shortfall (MES) method, which is a conditional probability modeling where payouts are given when the target variable exceeds the strike value. In contrast, Eze et al. (2020) used cluster analysis associating NDVI and weather variables with higher yield observations.

It is well known that climate variables present a certain degree of uncertainty when they are predicted. Therefore, this aspect needs to be considered when estimating losses caused by climate-related losses Smith and Matthews (2015). A stochastic approach based on Monte Carlo simulations is used in the literature to address the problem. A Monte Carlo simulation is the basis of the index modeling method applied by Gómez-Limón (2020), Guo et al. (2019), Kapsambelis et al. (2019), Gómez-Limón (2020), Mohor and Mendiondo (2017), and Rodríguez et al. (2021). The generation of synthetic weather time series enhances understanding the climate uncertainty in terms of confidence intervals. A summary of the risk pricing methods is described in Supplementary Information S1.

Econometric models provide values that guide decision-makers in understanding the price of the risk. However, it is fundamental to evaluate the risk reduction performance of index insurance. The simulation of cash flows allows an understanding of the hedging effectiveness of the insurance policy. Nonetheless, this efficiency depends on the point of reference adopted by the modeler.

The effectiveness problem arises when policyholders and insurance companies have different and often competing objectives. On the one hand, policyholders want to protect their assets at risk to prevent going out of business. On the other hand, insurance companies want to maximize profit to comply with the interests of their investors and shareholders. Since information asymmetry and moral hazards are allegedly minimized in the case of index insurance (Barnett et al., 2008; Mußhoff et al., 2018), the costs associated with moral hazards can be neglected from premium rates pricing.

The cash flow equation is a standard tool for evaluating the capital of companies and people. The simulation of cash flows using expected revenue and payouts as assets and premiums as liability for policyholders is used for evaluating the effectiveness of the index insurance policy (Bokusheva, 2018; Boyle et al., 2021; Kath et al., 2019; Martínez Salgueiro, 2019). For insurance companies, the cash flow changes the direction, i.e., premiums are considered assets, and payouts are a liability. This was used for calculating the loss ratio by (Mohor and Mendiondo, 2017).

Other authors have applied the utility theory to evaluate insurance policies. The utility theory accounts for the behavior and individual preferences in economic analysis and is based on several assumptions that apply to a group of individuals (Kahneman and Tversky, 1979). Some authors (Bucheli et al., 2021; Eze et al., 2020; Furuya et al., 2021; Vroege et al.) used the concept of risk-averse utility functions for policyholders, where the asset's utility at risk is concave or diminishing. Detailed information about the insurance policy evaluation methods is in Supplementary Information S2.

It is worth mentioning that either the method employed to estimate fair premium values and to tackle future risk increases due to climate change (CC) scenarios, the insurance market should consider some of the following points to reduce their weaknesses or uncertainty sources. First, as mentioned before, to calculate the premium value, it converges into a multi-objective problem.

The insured could contract a long-term insurance policy with an established premium. However, due to the CC uncertainty, some extreme events could not happen, and the insured paid too much for unnecessary coverage, resulting in more profit for the insurer. Nevertheless, the opposite case could happen. In that case, the insured pay less and extreme events happen, and the insurer does not have the liquidity to pay losses.

Second, a layered insurance scheme including private and public sectors (PP) (Keskitalo et al., 2014; Paudel et al., 2015) to cope with extreme losses. This means that when a certain threshold of loss is reached, a second partner will pay the difference in the indemnities. However, the definition of that threshold is another literature gap, similar to the strike value K.

Third, according to the spatial scale, a scheme of pool risk is preferable to reduce premium values which requires cooperation among the stakeholders. However, the main issue is to reach complete diversification of the portfolio (Porth et al., 2016). Fourth, this induces risk reduction proposals to increase resilience and promote adaptation within the sector. The latter could be reached through financial incentives such as premium discounts offered to stakeholders when they adopt some mitigation measure, as shown in (Hudson et al., 2016).

Finally, for a multi-hazard scheme, the schemes mentioned earlier should be calculated for each hazard. Nonetheless, premiums values will be different, and a weighted procedure will be required, such as done with the hazard frequency by Martínez Salgueiro (2019) and Guo et al. (2019) as mentioned in section 3.2.2.

## 3.3   Conceptual framework

Based on the results discussed, we present a conceptual framework for multi-hazard risk index-based insurance design de-
lineating various paths through which insurance policies can be developed. As shown in Figure 3, the index-based insurance design can be divided into three modules: (i) hazard identification; (ii) vulnerability assessment; and (iii) financial methods and risk pricing.

    Hazard identification is the process of analyzing and selecting the most critical threats and their respective indices. The vulnerability analysis refers to the process of selecting loss models and thresholds. Finally, the financial methods and risk
pricing refer to methods for estimating long-term loss expectations and risk premium rates.

    This framework (Figure 3)elicits paths for designing an insurance policy, and each step indicates a design option supported by the literature. Turvey et al. (2019), for example, followed a path A1-C1-D2-E1-F1, i.e., drought insurance (A1) with a static threshold (C1), a loss model based on losses projected by an index (D2), for single policyholders (E1) for many farmers in a region (F1). Sacchelli et al. (2018) provides another example, presenting a design path A2-B1-C1-D1-E1-F1, i.e., multi-
hazard risk insurance for wildfires and excessive rainfall (A2), considering the hypothesis of independent events (B1), with a static threshold (C1), using the index value to model losses (D1), and premium rates for a single policyholder (E1) without considering risk pooling (F1).

    **Hazard identification** The previous discussion demonstrated that the significant decisions in this step are whether to use a single (A1) or multi-hazard (A2). Single hazard risk is straightforward and should be reserved for situations when one hazard
is dominant in a region. However, research has shown that single-hazard hypothesis and multi-hazard risks might include independent, synergistic, and cascade events.

    According to (Gill and Malamud, 2014), independent hazards (B1) are events that can happen simultaneously in a region without any causal dependence. Synergistic hazards (B2) refer to a situation when the occurrence of a particular hazard increases the probability of the occurrence of another. Cascade hazards (B3) represent a situation when an event triggers the
occurrence of another event (i.e., excessive rainfall triggering landslides in a particular region).

    Alternative multi-hazard interaction hypotheses must be tested to depict weather and climate-related losses (Tilloy et al., 2019) and can influence the correct interpretation of loss modeling. The papers that addressed multi-hazard risks assumed independence between the events investigated; however, the combination of drivers and impacts of hazards contributes to risk analysis and is responsible for the occurrence of the most severe weather and climate-related impacts.(Zscheischler et al.,
2020). To overcome this challenge, Tilloy et al. (2019) presented methods for testing multi-hazard risk hypotheses, including copulas, classification algorithms, linear regression, and physical models. The importance of these models will be explored in the illustrative example.

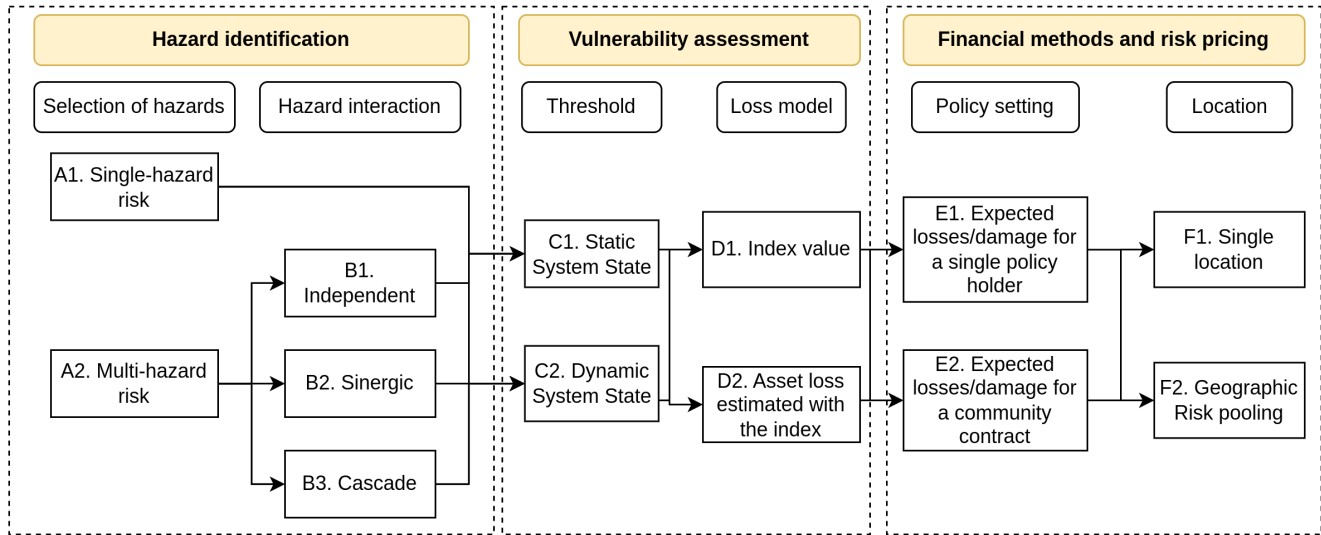

**Figure 3. The multi-hazard risks weather insurance design framework**. The framework illustrates the process of selecting and prioritizing hazards, defining index thresholds, modeling losses, and optimizing insurance risk premiums. The vulnerability assessment presents two types of systems: (a) Stationary System State, where both thresholds and hazards are stationary and represent an analysis based on observed historical information, and (b) Non-stationary System State, where both indices and hazards are non-stationary and reflect a combination of observed historical and projected data. The non-stationary system anticipates a potential increase in risk and optimizes risk premiums.

**Vulnerability analysis** The vulnerability analysis for insurance design is translated into threshold definition and loss model selection. Thresholds, strikes, and triggers are all terms for the pre-agreed-upon index value that triggers claim payments when reached or exceeded. In the literature, we identified loss models represented in terms of index value (D1) or losses estimated with index values (D2).

The rising frequency of extreme climate events has forced insurers to increase premium rates and threatens coverage availability. Losses and damages associated with extreme events have multiple drivers (Zscheischler et al., 2020), implying that losses have multiple thresholds and are associated with multiple variables. These thresholds vary with time and space (Hoek van Dijke et al., 2022).

As discussed in the vulnerability section, the selection of thresholds and consequential loss modeling consists in evaluating historical events. This creates a system we call Stationary System State (C1), characterized by fixed thresholds even when multiple hazards are considered. The second case is the Non-stationary System State (C2). The frequency and severity of hazards change over time, and the thresholds are dynamic, indicating either improvement or resilience deterioration.

We proposed a conceptual framework for considering multi-hazard risk analysis that allowed us to analyze the interaction between hazards and two types of vulnerability: static and dynamic resilience (Figure 4). Static resilience refers to a stationary state system. Most papers represent this case. Dynamic resilience refers to a Non-Stationary State System and considers changing hazard patterns and vulnerability thresholds. Considering a Non-Stationary State System helps to anticipate increas-

**Stationary System State ('static' resilience)**

– – – Stationary System's Vulnerability Thresholds

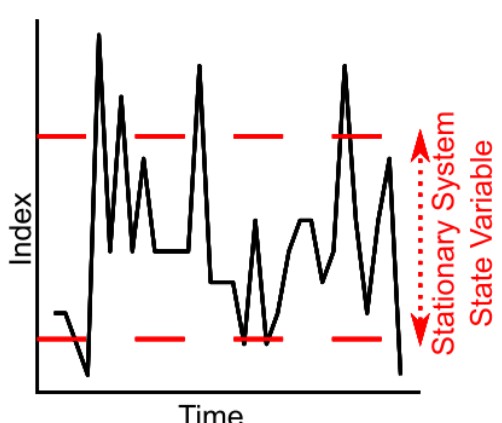

**Non-Stationary System State ('dynamic' resilience)**

· · · · Non-Stationary hazards
– – – Stationary System's vulnerability Thresholds

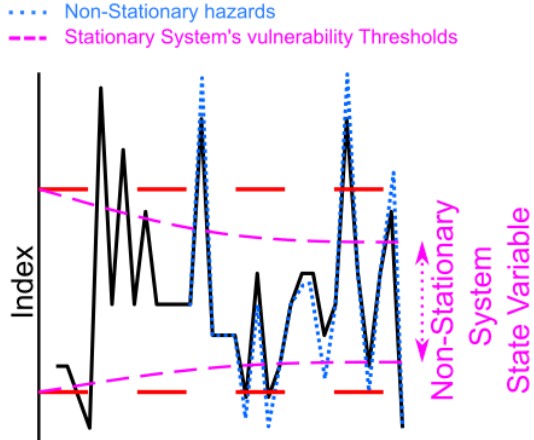

**Figure 4.** Graphical representation of static and dynamic state resilience systems

ing patterns of losses, therefore optimizing risk premiums to accelerate the adaptation and resilience of farmers against climate
change.

The dynamic threshold considers future scenarios that might assist in avoiding risk reassessment by anticipating and diluting potential severe climate shocks. Shifts in frequency and severity of extreme events are evaluated using the Representative Concentration Pathways (RCP) (Van Vuuren et al., 2011) indicating possible changes in risk exposure. The Shared Socioeconomic Pathways (SSP) (Riahi et al., 2017) will help us to understand risk in different vulnerability trajectories, i.e., increasing,
stationary, and decreasing resilience.

**Financial methods and risk pricing** Financial methods require defining parameters relevant to policy implementation, which depend on regional socioeconomic factors. Single contracts (E1) are focused on the individual policy and were the general option described in the literature review (Table 2). Community contracts (E2) are widely used for microinsurance contracts when smallholder farmers are associated with governmental institutions, associations, or companies to access affordable
insurance policies and long-term financial support against weather and climate-related threats (Platteau et al., 2017).

Both individual and community contracts can be tailored to a single region (F1), or different locations can be pooled (F2). In the derivative market, several locations can be insured in the same contract using the same weather index. This type of contract has been applied by retails (Štulec et al., 2019b) and is suited for farmers and companies with operations in more than one location.

## 3.4 Framework application: Illustrative example

An illustrative example was developed to demonstrate the multi-hazard risk path of the framework proposed. This encompasses all steps from problem definition and data collection to index calculation and loss evaluation for several cities and a specific crop. It is important to note that the methods utilized and the code created can be reused for different years, areas, countries, hazards, and crops.

This choice was motivated to prove the impact of selecting multi-hazard indices for designing weather index-based insurance for crop yields. Further studies must be done to link crop yield to other aspects of food security, such as transportation, storage, and retail.

In this example, we chose the 42 largest Soybean producing municipalities in the Brazilian state of Paraná for evaluation. According to Pereira et al. (2013), the study area is located in a region with low to very low geodiversity and low soil diversity. Fine-grained soil such as ultisols and latosols with high iron content are predominant in the region, followed by the presence of medium-grained soils such as cambisols (Bhering et al., 2009). These soils are generally classified as clayey and loamy, and their soil volumetric water content (VWC) tends to be high. According to Saxton and Rawls (2006), the Permanent Wilting Point (PWP) is, on average 22%, the field capacity (FC) is 37%, and the porosity/saturation is at 47%.

The production of soybeans represents an interesting object of study since it is an essential crop for oil and protein. Soybean has a significant economic impact in Brazil, wide geographical distribution, and a vulnerability to various hazards. Soybean crop yields in Paraná are mainly threatened by temperature variation, droughts, and excessive rainfall (da Silva et al., 2021).

We used 22 years of yield data of first-cycle soybean production from 1996/1997 to the 2019/2020 growing seasons. Crop data was retrieved from the official statistical yearbooks (Parana). The multi-hazard risk hypothesis was tested using the widely employed machine learning algorithm random forest (Breiman, 2001).

The conceptual framework illustrated in Figure 4 was applied to a case study for soybean production in South Brazil, following the methodology described in section 2.2. The main objective of this case study was to illustrate the main steps of the framework, focusing on multi-hazards risks (A2), testing the hypothesis of synergic interaction between hazards (B2), Stationary System State (C1), a loss estimated with index value (D2), single contracts (E1) and no risk pooling (F1). The following five key steps were used in the illustrative example:

**1st step - Hazard identification:** We selected thermal stress, drought, and excessive rainfall as the main threats to soybean production. The index selection was based on the indices found in the literature (Table 2) and the indices indicated by the CCl/-CLIVAR/JCOMM Expert Team (ET) on Climate Change Detection and Indices (ETCCDI) (Peterson et al., 2001). The index selection focused on finding simple indicators based solely on precipitation and temperature. After an extensive examination, the following indices were considered: maximum daily rainfall event over the growing season (pmax), 3-month Standardized Precipitation Index (SPI), number of days where daily precipitation is higher than the 90th percentile over the growing season (TX90p);

**2nd step - Definition of loss thresholds:** Crop losses were chosen as the target variable because they can be used as a proxy for the impact of extreme weather occurrences. The crop yields were detrended following the linear procedure used in Bucheli

et al. (2021) $\bar{y} = y_i + \beta \cdot (year_{end} - year_i)$, where $\bar{y}$ is the detrended crop yield series $y_i$ the raw crop yield data in the year i, $\beta$ is the linear regression coefficient of the equation $y_i = \alpha + \beta \cdot year_i$. The losses were then determined following the equation: $Loss = max(0, K - \bar{y}/K)$. The K variable is the crop yield threshold value. It can be understood as the threshold that divides unfavorable crop yields for farmers (values below K) and favorable crop yields (values above K);

**3rd step - Data clustering for evaluating the interaction between hazards:** The kmeans clustering method (MacQueen, 1967), a widely used clustering method, was implemented to understand the data better. The clustering was applied for four relevant variables: pmax, SPI, TX90p, and crop yield. The elbow method was used to define the optimal value of clusters (also referred to as hyperparameter $\kappa$). This is the most used method in the literature for defining $\kappa$. The method was implemented in R Environment using the package stats (R Core Team, 2022);

**4th step - Crop loss prediction modeling:** Several models were tested. However, two crop loss prediction models were chosen to demonstrate the importance of multi-hazard risk modeling, following a regression model and using the random forest algorithm: (i) Multi-hazard model M1 drought and thermal stressed using SPI and TX90p as inputs (M1(SPI,TX90p)); and (ii) Multi-hazard model M2 excessive precipitation and thermal stressed model using SPI, TX90p, and pmax as inputs (M2(SPI,TX90p,pmax)). The multi-hazard model M1 was trained and validated using data from clusters 2 and 4 and the multi-hazard model M2 was trained and validated using data from clusters 6. The standard cross-validation method was applied, following the best practices for machine learning workflows presented in the literature. The models were built using the R-package randomForest (Liaw and Wiener, 2002);

**5th step - Risk pricing:** The risk analysis is performed to determine pure risk premiums using stochastic methods. Historical burn analysis was performed on detrended crop yields to determine reference pure risk premium values. Then, a stochastic analysis of premiums for multi-hazard models M1(SPI,TX90p) and M2(SPI,TX90p) was determined considering P = E[Loss]. The expectation of loss E[Loss] was determined using the generation of 50 synthetic scenarios of weather data. The synthetic weather data was simulated using a multi-site multi-variable (daily precipitation and temperature ) weather generator method. The stochastic simulation was performed using a wavelet-based algorithm that allows multi-site simulation. The simulation was implemented in R-Environment with the package PRSim (Brunner et al., 2021).

The cluster analysis using climate indices and crop yield losses allowed us to interpret the multi-hazard nature of the historical loss events. We identified 6 clusters that are described in Table 4. Three clusters reasonably explained ca. 70% of soybean crop losses for the region and period studied. Cluster 2 represents years where losses were predominantly driven by precipitation deficit (Single hazard years).

Cluster 4 represents years where losses were driven by precipitation deficit and thermal stress (Multi-hazard year 1). Cluster 6 is associated with relatively normal years in SPI but with heavy rainfall events and higher temperatures (Multi-hazard year 2). The underlying structure of the other clusters (1, 3, and 5) can be related to other drivers of losses that were not considered in the present analysis.

The coupling of high temperatures and droughts have been a major cause of crop losses globally, and global warming is pointed to increased coupled thermal-moisture threats to food production (Lesk et al., 2021). On the other hand, excessive precipitation can increase soil moisture creating conditions for plant hypoxia, which means that plants have less access to

oxygen and have a reduction in their energetic status (Brandão and Sodek, 2009). Then, plants become more vulnerable to
other threats.

**Table 4.** Description of each cluster identified

| Cluster | Nº of obs | % of Losses | $SPI$ | $pmax$ | $TX90p$ |
|---------|-----------|-------------|-------|--------|---------|
| 1 | 389 | 14.4% | 0.702 | 42.4 | 9.55 |
| **2** | **153** | **86.3%** | **-0.941** | **36.1** | **13.4** |
| 3 | 162 | 19.1% | -0.320 | 37.9 | 22.4 |
| **4** | **106** | **96.2%** | **-1.340** | **39.8** | **33.4** |
| 5 | 110 | 27.3% | 1.390 | 76.5 | 11.1 |
| **6** | **46** | **95.7%** | **-0.357** | **70.9** | **22.5** |

The illustrative example highlights the multi-hazard nature effects of extreme weather events on crop yield losses. We used cluster analysis to identify what hazards were dominant each year that a crop loss event occurred. This results using a single hazard approach or assuming independence among hazards can be an oversimplification. The cluster method used for assessing multi-hazard events represent an option that can be used to visualize how to apply the proposed framework (3), which may improve decision-making in terms of index selection and vulnerability assessment. Improving hazard identification by categorizing historic data allows insurance designers having new insights in comparison to traditional methods, which conduct statistical analysis of past crop losses associated with a single hazard index.

We summarize the multi-hazard risk analysis for a specific city (Toledo), which is essential for soybean production and is severely impacted by extreme weather occurrences in Figure 5 it is possible to observe that: (i) Toledo had considerable losses on the multi-hazard periods (especially 2012); (ii) both models presented satisfying results for predicting crop losses on the different periods; (iii) the models identified different aspects of the data, implying that a model ensemble might produce the best results.; (iv) although multi-hazard model M2(SPI,pmax,TX90p) better suited the data (presenting a lower mean absolute error), multi-hazard model M1(SPI,TX90p) better predicted the worst year (2012), providing another evidence that an ensemble approach could present better results; (v) The sum of losses estimated with models M1 and M2 (Figure 5b) presented the lowest overall error in comparison to the observed crop loss, providing further evidence of the importance of using ensembles for predicting crop loss probability.

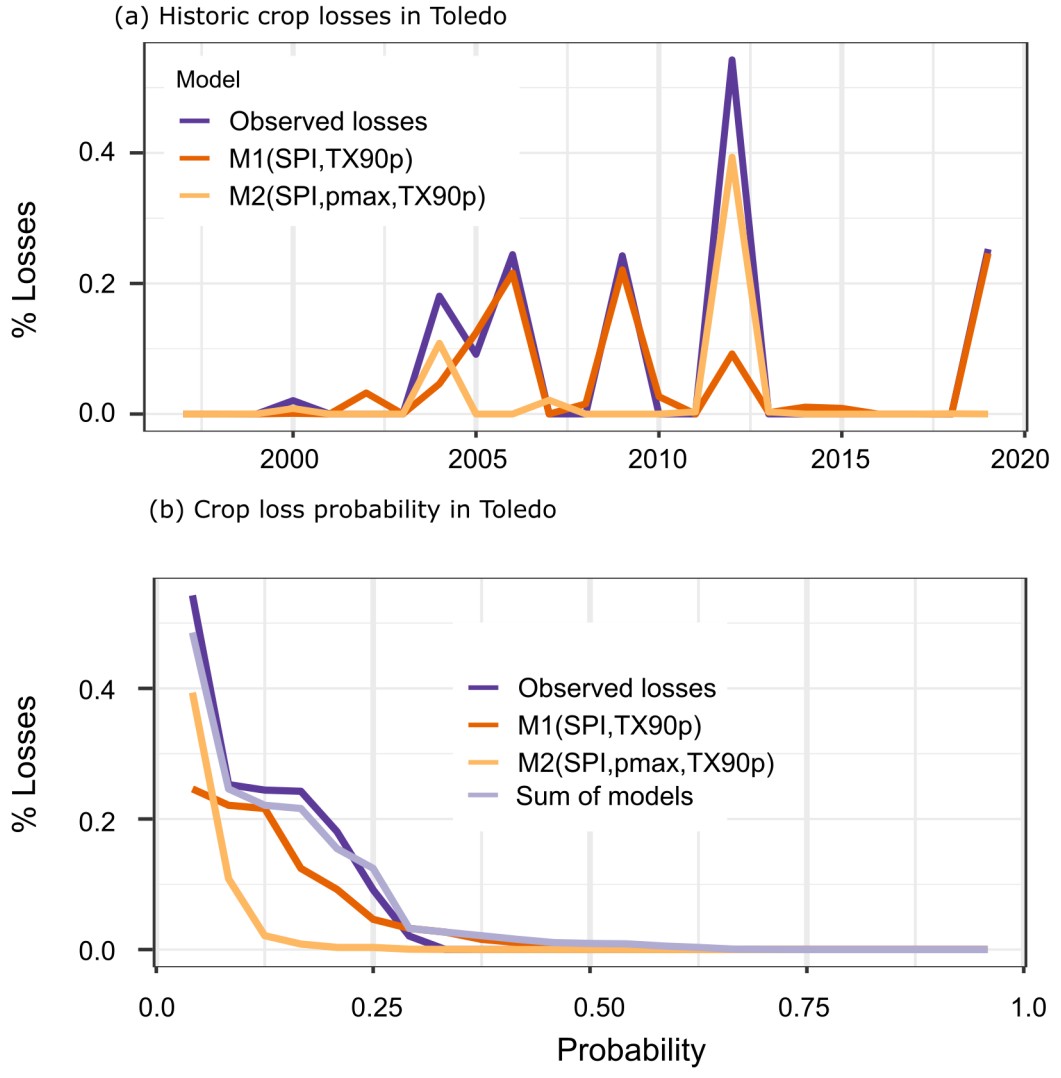

**Figure 5.** Risk analysis module applied to the city of Toledo for demonstrating the multi-hazard model M1(SPI,TX90p); multi-hazard model M2(SPI,pmax,TX90p), the sum of the two models; and observed crop losses in the studied period; (a) simulation of crop losses and (b) crop loss probability in the studied period including the sum of the losses estimated with two loss models

The study case conducted in this subsection illustrates one possible application of the framework, considering several analyses and visualizations that a stakeholder could use to understand better the impacts of extreme weather events over time on agricultural productivity, considering both the historical values and the crop loss probability. This information could improve insurance policy design and a better understanding of different regions' situations. Generating charts such as the ones illustrated in Figure 5 for multiple regions on a dashboard would allow for a better overview of the impacts of weather events on different crops and regions and be used to improve decision making.

**Table 5.** Summary of pure risk premiums in terms of percentage of expected crop yields

|  | Min | Mean | Max |
|---|---|---|---|
| Historical Burn Analysis | 2.745 % | 5.873 % | 9.722 % |
| Model 1 Synthetic scenario generation | 2.894 % | 3.253 % | 3.630 % |
| Model 2 Synthetic scenario generation | 0.519 % | 0.997 % | 2.506 % |

## 4    Conclusions

This study reviewed the development and design of index insurance focusing on multi-hazard risk analysis and food security. We summarized the primary hazard analysis and index calculation methods, loss modeling, and risk pricing. We observed that the lack of studies on multi-hazard risks is the central gap in the literature. Therefore, we proposed a conceptual framework with an illustrative example to give suggestions for future work in the field.

Drought was the most studied hazard, and cumulative precipitation index (CPI) was the most frequently used index in drought index insurance design. The literature review also presented other hazards, such as excessive precipitation, temperature variation, wind, and radiation. Since food security is a multifaceted concept, agricultural, hydrological and sustainable energy insurance were also evaluated.

The vulnerability analysis for insurance design was composed of the selection of a loss model selection and the definition of threshold values. Multi-hazard loss models were composed of generalized or separate additive models to calculate losses caused by different hazards when considering that the hazard occurrence was independent. Composite indices such as DOWKI or ambivalent indices such as CPI and SPI could capture excessive rainfall and droughts and are suitable for analyzing extreme conditions. Nonetheless, the reviewed papers did not fully explore other hypotheses of multi-hazard interaction, such as synergistic and cascade events.

Trigger values - the index value that triggers payouts - were attributed to both index values or losses estimated by the index. In the first case, thresholds were defined by quantile or a range of quantiles, e.g., 70 to 80th quantiles. The threshold value is dependent on how risk-averse the policyholder is. The case of losses estimated by the index was typical in crop insurance, where the threshold is a percentage of the expected crop yield for a given year.

The determination of risk premiums followed methods based on historical data evaluation. These methods are based on the assumption that historical data provides enough information to characterize regional risk. However, recent findings (Cremades et al., 2018) demonstrate that this approach might lead to an underestimation of future risk. In the in-depth analysis of the most relevant papers, we found burning rate, probabilistic fit, and index modeling as the most prevalent risk pricing models.

We proposed a conceptual framework with an illustrative example of multi-hazard index insurance design for soybean production in 42 municipalities in Parana state, Brazil. This application focused on categorizing multi-hazard events using a clustering technique based on the k-mean algorithm. Droughts and coupled thermal-moisture events were found in the study area. Two examples of multi-hazard events were detected by the clustering analysis, one is the combination of excessive rainfall

and high temperature and the other is the combination of droughts and high temperatures were detected by the clustering analysis.

The cluster model demonstrated that historic crop losses were divided into three groups: the first was precipitation deficit dominated, the second was precipitation deficit and high temperatures, and the third was excessive rainfall and high temperatures. Two different loss prediction models were trained with historic data separated according to the cluster analysis. This example illustrates that the problem of the mismatch between actual losses and losses predicted from the index insurance contract, also called basis risk, does not depend only on having enough historical records of loss events, but also on having enough understanding of what were the major drivers of loss events in the historical records. Future work must explore this effect and compare with actual yield.

Our paper demonstrates that, despite index insurance for food security has gained attention in the past years, there is still weaknesses and limitations that must be addressed in future work, e.g, a clear definition and analysis of multiple hazards instead of assuming single hazard risk; testing different hypotheses of the interaction between hazards, especially for coupled moisture-thermal events; evaluating how the multi-hazard risk selection affects basis risk; and analyzing the trade-offs between loss model accuracy and the policyholders willingness to pay.

*Author contributions.* Conception and design of the work: MRB. Data collection: MRB. Systematic Literature review: MRB, GCG, GJS, LCR, FARN, RFS. Discussion and analysis: MRB, GCG, GJS, LCR, FARN, RFS, EMM. Drafting the article: MRB. Critical review of the article: MRB, GCG, GJS, LCR, FARN, RFS, EMM, JAM, PAAM. Advisor: EMM

*Competing interests.* The authors declare that they have no known competing financial interests or personal relationships that could have appeared to influence the work reported in this paper.

*Acknowledgements.* This study was supported by the Coordenação de Aperfeiçoamento de Pessoal de Nível Superior do Brasil (CAPES) - Finance Code 001, and regular funding to post-graduate program in Hydraulics and Sanitation of University of Sao Paulo, São Carlos School of Engineering by the Brazilian National Council for Scientific and Technological Development (CNPq). Also, by the National Institute of Science and Technology for Climate Change Phase 2 (INCT-II) under the CNPq Grant 465501/2014-1, the São Paulo Research Support Foundation (FAPESP) Grant 2014/50848-9, the Center for Artificial Intelligence (C4AI-USP) under FAPESP grant #2019/07665-4, the IBM Corporation, and the CAPES Grant 16/2014, National Institute of Science and Technology (INCT) for the Fight Against Hunger under CNPq Grant 406774/2022-6.

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
