# Peer review of "Review article: Design and Evaluation of Weather Index Insurance for Multi-Hazard Resilience and Food Insecurity"

_EGUsphere, 2022_

## Author Comment (AC2)

*Response to Anonymous Referee #1*

**General comments**

The subject of the article 'Review article: Design and Evaluation of Weather Index Insurance for Multi-Hazard Resilience and Food Insecurity' is interesting. The review methodology is generally transparent. However, many issues need to be addressed. The overall presentation of results, discussion, and conclusions are very rough and often confusing. English is poor, with many grammatical and conceptual errors. The article requires professional editing to improve both language and readability. A major revision is needed before it can be accepted for publication in this Journal.

Response: We thank the Anonymous Referee #1 for his/her comments on our manuscript. We will address each of the suggestions in order to improve it, so it can be accepted for publication.

**Specific comments**

Terminology is an issue. E.g. 'sustainable insurance' is used all over the article and implies insurance for sustainable energy production. However, the expression is not correct, as it actually means insurance that is sustainable. Or, the abstract's phrase 'preferred systematic reviews' probably refers to a specific review method (I assume PRISMA), but it is not evident at all the way it is used, i.e., as a common adjective.

Response: We agree with the reviewer. The terminology "sustainable insurance" will be changed to "sustainable energy production insurance". The second suggestion will be clarified in the abstract and the text.

The title promises a 'Design and Evaluation of Weather Index Insurance for Multi-Hazard Resilience and Food Insecurity'. However, we learn that the studied papers lack a multi-hazard approach, while the authors only refer to some examples that need to be studied. They do not propose any specific method to address this issue.

Response: In fact our first idea was to use the conceptual framework presented in Figure 4 to address this issue, however, we believe that, based on the review, we need to create a sub-section to better discuss the conceptual framework.

As in the title, the abstract also states that 'This paper aims to provide a state-of-art weather index insurance design.' This sounds quite promising. However, there is only 1 line dedicated to the design of Weather Index Insurance, in 3.2.3, L383-384: 'Finally, we present a conceptual framework derived from the literature review representing the weather index insurance design process (Figure 4).' Are this figure and the enclosed proposed design/process given as a result of the most used methods within the reviewed literature? Is this proposed design a synthesis that presents something new? How is multi-hazard resilience addressed in particular through this design? Finally, I would suggest a separate sub-section to present and discuss the proposed design, to support its effectiveness based on a case study, to highlight its effectiveness on multi-hazard index insurance, and, further, on resilience and food insecurity. Otherwise, the title and the abstract are not well connected to the results.

Response: As we stated in the previous response, we appreciate the suggestion of creating a new sub-section to describe the conceptual framework in more detail. We are preparing a case study to illustrate our concept and show how to address the questions raised by the reviewers mathematically.

3.2.2: the entire section requires thorough English editing and better development. It seems very draft, full of sentences without verbs, confusing the reader. Also, the sub-sections 3.2.1-3.2.2 include some same comments and results. Several consecutive paragraphs should be merged as they deal with the same subject (see for example, L346-356, and many others…)

Response: We will address the language and editing problem as suggested.

L313: Table S3 presents loss models. Vulnerability as a title is not mentioned. Either it is a misconception, or it needs clarification. Actually, the models are called vulnerability models; however, their objective seems to be to estimate the loss and not the vulnerability.

Response: We thank the reviewer for this comment, we believe that we need to reflect the table mentioned in the supplementary material and make the proper clarification, so the difference between loss models and vulnerability models is fully explained.

**Conclusions**

The conclusions summarize the results but are written in a very rough form. L420: which gaps were observed? This should be highlighted in conclusions and complemented by suggestions.

Response: We thank the reviewer for this comment, we will revise the text and make sure the conclusions are better rendered and the gaps better explained.

**Technical comments**

Response: We will revise all the technical comments made by the reviewer. We thank you for the thorough revision and attention to details. We believe that this will make our manuscript stronger and in shape for publication.

---

## Author Comment (AC3)

**Response to Anonymous Referee #2**

With the recent emergence of index insurance papers and the myriad approaches and hazards available, a review paper is warranted. The title indicates a focus on multi-hazards and food insecurity, however the manuscript moves far beyond these topics. Significant amounts of information spanning hazard type, variables, approaches, etc., are presented, which turns out to be overwhelming instead of comprehensive. Arguably the manuscript would benefit from a greater focus; presently it has too much breadth and lacks depth. In its current form, it is unclear how readers can benefit from this contribution. A reformatting of the presentation with a clearer emphasis and concrete take-aways could lead to a valuable contribution. Additional suggestions are presented below:

**Response:** We thank the Anonymous Referee #2 for his/her comments on our manuscript. We will improve our manuscript by focusing on food security and reformatting the manuscript. Moreover, we will address each of the suggestions in order to improve it, so it can be accepted for publication.

Questions at the end of the Introduction section are very relevant, but also very broad. Perhaps focusing specifically on multi-hazards and food insecurity can help to focus the questions further (and presumably the manuscript as a whole.) Related, it's not well motivated why other sectors (e.g. energy) are included here. Unless they specifically relate to energy used in food production? Or tradeoffs between hydropower and allocations to agriculture? More specificity is strongly suggested to tighten the analysis and findings.

**Response:** Thank for your comment. We believe that our work should undergo substantial improvements if we focus on multi-hazard and food insecurity. We have enough material to focus on this aspect. We understand that trade-offs between other sectors it's a great suggestion for future work, so we will discuss it briefly in the conclusions and future work.

Table 1: Are these Themes developed by authors or follow a commonly accepted methodology? More description is required on what centrality and density really mean and how best to be interpreted. Also, what is the reference for the numerical scale? 0-10? Finally, some clusters are described in numerous Themes. For example, Basic includes four clusters in the text, but only two in the figure.

**Response:** We thank you for your comment. We obtained these themes followed the keyword clustering algorithm proposed by Cobo et al. (2011) and we determined it using the package bibliometrix in R Environment. Cobo et al. based their metrics in the concept of centrality and density of co-word analysis developed by Callon et al (1991). The method is a practical application of the Fuzzy Set Theory Field and does not have a specific scale. The values are relative do the number of papers analyzed, number of citation in the documents and the number of clusters generated by the co-word analysis. We will describe more clearly how these metrics – density and centrality – are calculated and how to better interpret them.

The Hazard Identification section feels like a really long list, and it's unclear what the goal is in this section. Is it to essentially list the papers that go with each hazard? That could be doable in a large Table (perhaps appendix.) The descriptions of the hazards are basic enough that most readers should be familiar (and arguably aren't learning new information), perhaps with the exception of SPI or similar, but even then, most are likely to already know. It's of course necessary to identify the top hazards (e.g. Fig 3), but the text in its current form does not add much. The authors are encouraged to consider either simplifying (e.g. a Table as suggested) or taking a deeper dive into the details of hazard aspects identified and targeted in each paper.

**Response:** We thank you for your comment. We agree with your comment and we believe that the manuscript will benefit by reducing the description of the hazard calculation in a table and proposing an in-depth discussion of index selection, impacts on design and decision-making. We learned from the literature review that the risk communication is much appreciated and farmers tend are willing to purchase a policy when they understand risk.

The Vulnerability Analysis section appears to be a mix of discussing assets, variables, and modeling approaches. Perhaps the authors could consider an alternative presentation approach such as combining parts with the Hazard Identification section (e.g. have subheadings by hazard type that also includes assets at risk and relevant variables.) This could partially address the comment above. The description of modeling approaches does not appear to fit in this section, and is probably a stand-alone section.

**Response:** The vulnerability analysis is an important step for our review and conceptual framework. We agree with Referee #2' comments. We will clarify our choice by explaining what vulnerability concept we have adopted in our work. Therefore, some of the language, e.g., the words 'assets' and 'assets at risk' will be substituted for a formal and clear definition of vulnerability and then how we approached the literature review.

While the Financial Methods section relays the large number of design approaches, target outcomes, etc. It is unclear what the reader is supposed to take away from this section. Simply that there are many types? Or if a particular type sounds most appealing or relevant to their needs, then they can refer to the papers cited? Certainly some aspects are appealing, such as those that describe the pros and cons of a particular approach. The authors are encouraged to include more pros and cons to give the reader a more firm understanding and perhaps guidance for their own work.

**Response:** We thank you for your comment. We will improve the discussion on what are advantages and disadvantages of the models and potential implications in decision-making. We will make sure that at the end of the section the readers are able to select the methods according to different multi-hazard scenarios.

For the Conclusions, it may be a stretch to claim that a large number of papers (ag and crop insurance) leads to a 'high impact on index insurance'. It does mean the topic is perhaps more well studied than others, but I believe the impact is still

very small. Also, I'm not sure this manuscript really points to the 'gaps in the field', and if it does, then this needs to come across much more strongly. Conclusions that point directly to multi-hazard and food insecurity should be front and center in this section. I suggest expanding this section.

**Response:** We thank you for your comment. We agree with the referee comment. We need to clarify that the impact was calculated using Cobo's metrics of density and centrality and it means that crop insurance theme has a high impact in the literature that we collected in the systematic review. However, overall, index insurance has a lot of potential and we agree that field in general has a lot of room for improvements. We found two main gaps in the field. The first one is that only a few studies approached multi-hazard insurance design and second, the ones that approached did not further described how to perform an exploratory multi-risk hazard analysis for index insurance. This is particularly challenging in countries such as Brazil, which are vulnerable to multiple climate hazards. We will expand this section and will shed more light on food insecurity and multi-hazard.

The Reviewer acknowledges that review papers are challenging to write. The authors are encouraged to highlight their motivation for assembling the manuscript, more clearly focus the topics, and articulate precisely what they want readers to glean from this contribution.

**Response:** We once again thank the reviewer for carefully reading our manuscript and being so insightful. We intend to address all the questions, particularly; we will focus more in the implications of multi-hazard risks in food security. We believe that manuscript will largely benefit from the revision.

**References:**

Cobo, M., López-Herrera, A., Herrera-Viedma, E., and Herrera, F.: An approach for detecting, quantifying, and visualizing the evolution of a research field: A practical application to the Fuzzy Sets Theory field, Journal of Informetrics, 5, 146–166, https://doi.org/10.1016/j.joi.2010.10.002, 2011.

Callon, M., Courtial, J. P., & Laville, F.: Co-word analysis as a tool for describing the network of interactions between basic and technological research—the case of polymer chemistry, Scientometrics, 22, 155–205, 1991

---

## Author Response (AR1)

Dr. Vassiliki Kotroni

Editor

*Natural Hazard and Earth System Sciences*

October 25$^{th}$, 2022

Re: Review of manuscript **egusphere-2022-498**

Dear Dr. Vassiliki Kotroni,

We are very grateful for each comment and suggestion made by the two referees, which turned our manuscript more powerful.

Our manuscript presented a systematic review of weather insurance design for food security considering multi-hazard risks. We observed that the recent literature provides little examples of multi-hazard risk analysis in weather insurance design.

Following the comments from the referees, we rendered the systematic review to highlight key findings and we expanded the section about our conceptual framework making it a separate sub-subsections and providing a study case for soybean production in Brazil.

The study case employs the methods for index calculation, loss modeling and premium pricing we found and the literature review and adds our view on how multi-hazard risk insurance should be designed when considering water deficit, heavy rainfall and extreme temperature. We applied kmean clustering to define multi-hazard scenarios and used random forest algorithms to predict losses for each scenario identified.

We suggest that this framework can be applied to more segments of the food supply chain such as transportation, storage and retail. Moreover, we identified emmering topics of weather index insurance such as hydrological and sustainable energy insurance.

We also made a very detailed revision throughout the manuscript to avoid English grammar and typos issues.

Additionally, we answered carefully point-by-point each comment made by the two anonymous Referees. Please see it below.

For this iteration of our manuscript submission we have the help of two co-authors: Prof. Dr. Patricia Marques and Dr. José Marengo. Their participation in the critical review of the manuscript was critical for us to adequate the old to the suggestions made by the referees.

We hope that the revisions in the manuscript accompanied by this supporting letter will now meet the requirements for publication in *Natural Hazard and Earth System Sciences*

Thank you for your consideration.

On behalf of all authors,

Marcos R. Benso

**Anonymous Referee #1**

**General comments**

The subject of the article 'Review article: Design and Evaluation of Weather Index Insurance for Multi-Hazard Resilience and Food Insecurity' is interesting. The review methodology is generally transparent. However, many issues need to be addressed. The overall presentation of results, discussion, and conclusions is very rough and often confusing. English is poor, with many grammatical and conceptual errors. The article requires professional editing to improve both language and readability. A major revision is needed before it can be accepted for publication in this Journal.
Response: We are very grateful for the comments and suggestions made. We reformulated the presentation of the methodology, results and discussion. In addition, we added a case study to elucidate the framework proposal. Regarding the english language editing, we made a detailed revision so it can be accepted for publication

**Specific comments**

Terminology is an issue. E.g. 'sustainable insurance' is used all over the article and implies insurance for sustainable energy production. However, the expression is not correct, as it actually means insurance that is sustainable. Or, the abstract's phrase 'preferred systematic reviews' probably refers to a specific review method (I assume PRISMA), but it is not evident at all the way it is used, i.e., as a common adjective.
Response: We agree with the reviewer. The terminology "sustainable insurance" was changed to "sustainable energy production insurance". We reformulated the abstract with extra attention to the reviewer's comment. The sentence was written as the following: *"There is a growing attention in the literature for the topic of weather index insurance and resilience of climate-sensitive sectors such as food production. [...]"*

The title promises a 'Design and Evaluation of Weather Index Insurance for Multi-Hazard Resilience and Food Insecurity'. However, we learn that the studied papers lack a multi-hazard approach, while the authors only refer to some examples that need to be studied. They do not propose any specific method to address this issue.

Response: Thank you for your comment. We agree about the lack of a multi-hazard approach and gave a better description of it in developing a study case. The study's methodology was described in the methodology section 2.2 (L132-169) and the results were presented in section 3.3.2 (L423-471). The study case allowed us to better discuss the conceptual framework and tackle the issue raised by the reviewer.

As in the title, the abstract also states that 'This paper aims to provide a state-of-art weather index insurance design.'This sounds quite promising. However, there is only 1 line dedicated to the design of Weather Index Insurance, in 3.2.3, L383-384: 'Finally, we present a conceptual framework derived from the literature review representing the weather index insurance design process (Figure 4).' Are this figure and the enclosed proposed design/process given as a result of the most used methods within the reviewed literature? Is this proposed design a synthesis that presents something new? How is multi-hazard resilience addressed in particular through this design? Finally, I would suggest a separate sub-section to present and discuss the proposed design, to support its effectiveness based on a case study, to highlight its effectiveness on multi-hazard index insurance, and, further, on resilience and food insecurity. Otherwise, the title and the abstract are not well connected to the results.

Response: As we stated in the previous response, we appreciate the suggestion and create a new subsection. The subsection is "**3.3 Conceptual framework and study case**" including the sub-subsections "**3.3.1 Conceptual framework**" (L403-421) presenting a synthesis of a new idea to address the multi-hazard problem we identified in the literature review. This includes the addition of common methods we found within the reviewed literature; and "**3.3.2 Study Case**" (L423-471) where we present Figures 5, 6 and 7 demonstrating the process of selecting multi-hazard risks using the well-known machine learning clustering method k-means. We also describe tables 3 and 4 to explore the study case results . We reformulated Figure 4 to better demonstrate how to address multi-hazard resilience and to incorporate the concepts of static and dynamic resilience.  In addition, we reformulated the abstract to be consistent with the content of the article.

3.2.2: the entire section requires thorough English editing and better development. It seems very draft, full of sentences without verbs, confusing the reader. Also, the sub-sections 3.2.1-3.2.2 include some same comments and results. Several consecutive paragraphs should be merged as they deal with the same subject (see for example, L346-356, and many others…)

Response: Thank you for all your suggestions and comments. We made a detailed revision in order to avoid any language problems. The section 3.2 was reformulated as we previously stated and divided into two sub-subsections.

L313: Table S3 presents loss models. Vulnerability as a title is not mentioned. Either it is a misconception, or it needs clarification. Actually, the models are called

vulnerability models; however, their objective seems to be to estimate the loss and not the vulnerability.

Response: We thank the reviewer for this comment. We clarify throughout the text and supplementary material the difference between loss models and vulnerability models. In addition, we update the table S3.

**Conclusions**

The conclusions summarize the results but are written in a very rough form. L420: which gaps were observed? This should be highlighted in conclusions and complemented by suggestions.

Response: This is an important point and thanks you for pointing it out. We reformulated the conclusions, included explicitly the gaps, and made recommendations for future studies.

**Technical comments**

Abstract

L1: what do you mean by 'preferred' systematic reviews? PRISMA is not implied here; thus, it sounds like an awkward adjective.

Response: We replaced this statement for the following: "Weather index insurance has gained growing attention in the literature. Several approaches have been employed to determine indices, model losses and calculate fair premium rates, however, little attention has been given to define generalized approach that analyzes multi-hazard risk for insurance design."

L5: please correct: to 'the' present

Response: Corrected!

L10: This sentence needs grammar correction.

Response: We corrected this statement for the following: "Despite the great focus on food security, emerging fields such as hydrological and sustainable energy were found promissory for index insurance and will require further systematization".

Introduction

L18. References should be put in parenthesis

Response: Corrected L20.

L27: please delete the second 'the'

Response: Corrected L30.

L29: consider specifying that these are amounts for premiums per capita

Response: We added the "hab" symbol to make it clear: *"In one hand, the premiums per capita (hab) of the US and Canada were 7,270 USD/hab, much higher than the world average of 809 USD/hab and the Eurozone average of 2,723 USD/hab. On the other hand, in Latin America and the Caribbean, and emerging Europe and Asia presented 203, 159 and 215 USD/hab respectively. The numbers were much lower in Africa and the Emerging Middle East, representing 45 and 93 USD/hab"*

L34: please rewrite the sentence as a verb seems missing
Response: We corrected the sentence: *"The area-yield insurance model **was adopted** in the US in the early 90s, dividing agricultural areas in the crop domain into Group Risk Plans (GRP)"*

L58: please correct the reference presentation inside the sentence
Response: Corrected L61.

Methodology

L100. Please correct: 1192 studies were selected
Response: Corrected L101

Figure 2: (a) please include a legend for series
Response: Corrected.

Results

L161: Please consider rewriting the sentence to make sense
Response: The paragraph was reformulated and during this process, we rewrote the sentence, please see L223-225.

L163-165: these sentences have to be corrected for grammar and language. The parenthesis is awkward. Which study are the authors referring to??
Response: The sentence was removed and the paragraph was rewritten.

L170: please delete 'and' before 'wind…'
Response: Corrected L253.

Figure 3. Please use consistent fonts/colors…I don't understand the meaning of the box sizes and positions.
Response: We corrected the colors and fonts using a color blind and print safe divergent color sequence. The meaning of the boxes were added to the legend. The bigger the box size the more papers the index was used.

L198-199: please rewrite. The authors 'concluded' or just 'suggested'?

Response: The authors concluded since they performed a statistical analysis of the results.

L201: please put space before the parenthesis
Response: Corrected.

L218-219: please rewrite as the indices are not grammatically connected to the rest of the sentence.
Response: Corrected L308-310

L257: please consider rewriting; the sentence is not clear in what concerns the extreme of the distribution. May a verb be missing?
Response: We rewrote this paragraph L320-327

L260: A conjunction is missing (and or while…??)
Response: We rewrote this paragraph L320-327

L263: please correct: 'evaluated'
Response: We rewrote this paragraph L320-327

L264: please correct: 'by the increase in..'
Response: We rewrote this paragraph L320-327

L271: please rephrase. 'We observed an emerging topic affecting sustainability with a focus on sustainable energy generation' does not make sense.
Response: We are sorry if this sentence was not very clear. We made some improvements. Please see L328-337.

L266-270: this has been already said in previous sections
Response: Thank you for your suggestion. We removed the duplicated information. L320-327.

L271-276: I don't understand the difference between this paragraph and the last one in 3.2.1.
Response: We reformulated the paragraph, please see Please see L328-337.

L277-280: This paragraph is not well written. E.g., moral hazards cannot be neglected, but rather the opposite. They seem to be considered. Also, 'Basis risk, and it implies…' is grammatically incorrect and the sentence makes no sense.
Response: We reformulated the section 3.3.2 Vulnerability Analysis and we decided to remove this paragraph.

L283-284: Another sentence that is grammatically incorrect…Please rephrase.
Response: We reformulated the section 3.3.2 Vulnerability Analysis and we decided to remove this paragraph.

L293: what is meant by '..and its interaction'? interaction with what?
Response: We reformulated the section 3.3.2 Vulnerability Analysis and we decided to remove this paragraph.

L299: I am not sure I understand well this sentence. What method/tool was used and how were the hazards included as independent?
Response: We reformulated the section 3.3.2 Vulnerability Analysis and we decided to remove this paragraph.

L305-307: please clarify: 'this variable should be considered to improve the model's ability…' Which variable? Why 'should' it be considered'? based on which evidence?
Response: We reformulated the section 3.3.2 Vulnerability Analysis and we decided to remove this paragraph.

L311: consider putting a comma before 'giving'. Oherwise it does not make sense.
Response: We reformulated the section 3.2.2 Vulnerability Analysis and we decided to remove this paragraph.

L320: streamflow 'is' low
Response: sub-subsection 3.2.3 Financial methods and risk pricing was reformulated and this sentence was removed.

L321: which 'extreme condition'?
Response: sub-subsection 3.2.3 Financial methods and risk pricing was reformulated and this sentence was removed.

L322: 'sustainable insurance' sounds not appropriate. Do the authors mean 'sustainable energy production'??
Response: We updated it for "sustainable energy production insurance".

L341: what is meant by 'full information'?
Response: sub-subsection 3.2.3 Financial methods and risk pricing was reformulated and this sentence was removed.

L342: please correct 'They are calculated historical data…'
Response: sub-subsection 3.2.3 Financial methods and risk pricing was reformulated and this sentence was removed.

L341-345: this paragraph is confusing. Mixing non-index and index insurance as if this is the first time index insurance is mentioned, while this is the main subject of this review paper.

Response: The sub-subsection 3.2.3 Financial methods and risk pricing was reformulated and this paragraph was reformulated.

L347-348: grammatically incorrect sentence.

Response: The sub-subsection 3.2.3 Financial methods and risk pricing was reformulated and this sentence was reformulated.

Fig4: please correct : (c) amount. Also correct the figure caption: 'fnancial risk pricing (e) and (e) and (f)'

Response: The figure was reformulated.

L386: note that Table 2 should be in parenthesis, or included correctly in the sentence.

Response: We are sorry about this formatting error.

L398: WTP, what does it mean? Please clarify

Response: Thank you for noticing it. The acronym was explained. WTP means willingness-to-pay.

Paragraph L397-409: the entire paragraph has grammar mistakes, verbs missing, language issues. It requires careful editing

Response: The sub-subsection 3.2.3 Financial methods and risk pricing was reformulated and this sentence was reformulated.

**Anonymous Referee #2**

With the recent emergence of index insurance papers and the myriad approaches and hazards available, a review paper is warranted.  The title indicates a focus on multi-hazards and food insecurity, however the manuscript moves far beyond these topics.  Significant amounts of information spanning hazard type, variables, approaches, etc., are presented, which turns out to be overwhelming instead of comprehensive.  Arguably the manuscript would benefit from a greater focus; presently it has too much breadth and lacks depth.  In its current form, it is unclear how readers can benefit from this contribution.  A reformatting of the presentation with a clearer emphasis and concrete take-aways could lead to a valuable contribution.  Additional suggestions are presented below:

Response: We would like to thank you for your kind words in support of our study and for suggesting improvements to the manuscript. We are going to address each of your comments carefully.

Questions at the end of the Introduction section are very relevant, but also very broad. Perhaps focusing specifically on multi-hazards and food insecurity can help to focus the questions further (and presumably the manuscript as a whole.) Related, it's not well motivated why other sectors (e.g. energy) are included here. Unless they specifically relate to energy used in food production? Or tradeoffs between hydropower and allocations to agriculture? More specificity is strongly suggested to tighten the analysis and findings.

Response: Thank you for your comment. We believe that our work should undergo substantial improvements if we focus on multi-hazard and food security. We have enough material to focus on this aspect. We understand that trade-offs between other sectors it's a great suggestion for future work, so we discussed it briefly in the conclusions and future work.

Table 1: Are these Themes developed by authors or follow a commonly accepted methodology? More description is required on what centrality and density really mean and how best to be interpreted. Also, what is the reference for the numerical scale? 0-10? Finally, some clusters are described in numerous Themes. For example, Basic includes four clusters in the text, but only two in the figure.

Response: We thank you for your comment. We obtained these themes following the keyword clustering algorithm proposed by Cobo et al. (2011) and we determined it using the package bibliometrix in R Environment. Cobo et al. based their metrics on the concept of centrality and density of co-word analysis developed by Callon et al (1991). The method is a practical application of the Fuzzy Set Theory Field and does not have a specific scale. The values are relative to the number of papers analyzed, the number of citations in the documents, and the number of clusters generated by the co-word analysis. We described more clearly how these metrics – density and centrality – are calculated and how to interpret them in L.

The Hazard Identification section feels like a really long list, and it's unclear what the goal is in this section. Is it to essentially list the papers that go with each hazard? That could be doable in a large Table (perhaps appendix.) The descriptions of the hazards are basic enough that most readers should be familiar (and arguably aren't learning new information), perhaps with the exception of SPI or similar, but even then, most are likely to already know. It's of course necessary to identify the top hazards (e.g. Fig 3), but the text in its current form does not add much. The authors are encouraged to consider either simplifying (e.g. a Table as suggested) or taking a deeper dive into the details of hazard aspects identified and targeted in each paper.

Response: We thank you for your comment. We reformulated the entire section and described the hazard calculation in Table 2 in addition to an in-depth discussion of index selection and impacts on design and decision-making. We learned from the literature review that risk communication is much appreciated and farmers are more willing to purchase a policy when they understand risk.

The Vulnerability Analysis section appears to be a mix of discussing assets, variables, and modeling approaches. Perhaps the authors could consider an alternative presentation approach such as combining parts with the Hazard Identification section (e.g. have subheadings by hazard type that also includes assets at risk and relevant variables.) This could partially address the comment above. The description of modeling approaches does not appear to fit in this section, and is probably a stand-alone section.

Response: We appreciate your suggestion. We explained the concepts we adopted in the paper in L111-131. We believe that giving this context improves the understanding of our discussion. Furthermore, we addressed this comment and the previous by rewriting both subsections 3.2.1 Hazard Assessment and 3.2.2 Vulnerability analysis.

While the Financial Methods section relays the large number of design approaches, target outcomes, etc. It is unclear what the reader is supposed to take away from this section. Simply that there are many types? Or if a particular type sounds most appealing or relevant to their needs, then they can refer to the papers cited? Certainly some aspects are appealing, such as those that describe the pros and cons of a particular approach. The authors are encouraged to include more pros and cons to give the reader a more firm understanding and perhaps guidance for their own work.

Response: We thank you for your comment. We improved the discussion section and discussed the advantages and disadvantages of the models and potential implications in decision-making. Please see L353-401.

For the Conclusions, it may be a stretch to claim that a large number of papers (ag and crop insurance) leads to a 'high impact on index insurance'. It does mean the topic is perhaps more well studied than others, but I believe the impact is still very small. Also, I'm not sure this manuscript really points to the 'gaps in the field', and if it does, then this needs to come across much more strongly. Conclusions that point directly to multi-hazard and food insecurity should be front and center in this section. I suggest expanding this section.

Response: We thank you for your comment and agree with it. We reformulated the entire conclusion section with special attention to the gaps in the field. In addition, we expand the section to shed more light on food security and multi-hazard.

The Reviewer acknowledges that review papers are challenging to write. The authors are encouraged to highlight their motivation for assembling the manuscript, more clearly focus the topics, and articulate precisely what they want readers to glean from this contribution.

Response: We once again thank the reviewer for carefully reading our manuscript and being so insightful. We try our best to address all the questions, particularly; we will focus more on the implications of multi-hazard risks in food security.

---

## Referee Report (RR1)

The article proposes a literature review of multi-hazard weather-index insurance for agriculture, specifically for crops. The paper would like to identify which indices are used to assess and monitor extreme weather events, what functions and methods are used to determine the vulnerability of food production to multi-hazard events and how to compute risk premiums. The authors applied the PRISMA protocol to identify 34 studies on the selected topic. In addition, they propose a conceptual framework to solve the problem of multi-hazard risk and minimizing the premiums (lines 404-406). The conceptual framework is applied to the production of soybean in Brazil.

The topic of multi-hazard index insurance deserves for sure further attention, as underlined by the authors, thus a review of studies addressing multi-hazard parametric insurance for crops is a valuable contribution for the scientific community.

However, my feeling is that the paper in its current form is a little bit confusing for the reader. First of all, the authors state in the abstract and in the introduction that their "primary focus is considering a multi-hazard approach and selecting studies in food security" (line 5) and they would like to answer the question "What functions and methods are used to assess the vulnerability of food production to extreme weather events?" (lines 74-75). However, inside the paper there are continuous reminder to insurance for renewable energy production and hydrological risk and it is not clear if the two aspects are related with food security or not. If this is the case, the authors should explain the connection better.

Secondly, the paper seems to be divided into two distinct parts: a first one dealing with the literature review and a second one explaining the conceptual framework used to solve the problem of multi-hazard risk and minimizing the premiums.

Proposing a conceptual framework and applying it to a specific case study inside a literature review article sounds strange since usually a literature review explores the studies on a specific topic and provides information on gaps, shortcomings and future research areas for that specific topic.

In addition, both the parts, the literature review and the proposed conceptual framework, are not deepened enough.

In the sections 3.2.1, 3.2.2 and 3.2.3 (Hazard assessment, vulnerability analysis and financial methods and risk pricing) the discussion is too simple. In section 3.2.1 I would have expected a discussion on the advantages and disadvantages of the application of the described indices in parametric insurance and an insight on the indices adopted for multi-hazard risk assessment. In section 3.2.2 there is no discussion on the pros and cons of the described methodologies used to determine crop vulnerability to multi-hazard events. Finally, in section 3.2.3 again a discussion on the strengths and weaknesses of the methods proposed in the literature to determine fair premiums is lacking. Therefore, I suggest to improve the sections.

Basis risk, which is a crucial point for the effectiveness of index-based insurance is often mentioned, but details on the basis risk of the insurance programs designed in the reviewed studies are not proposed to the reader. This point should be better investigated by the authors, including in the text some considerations on basis risk proposed in the reviewed studies.

Finally, in the conclusion section I would expect to find the answers to the three questions raised in the introduction:

1) What indices are used to assess and monitor extreme weather events?

2) What functions and methods are used to assess the vulnerability of food production to extreme weather events?

3) How to determine risk premiums?

Instead, the first part of the conclusions underlines the lack of studies on index insurance tailored to Latin America, while the second part describes the results obtained by applying the conceptual framework to the case study area. I recommend the authors to include in the conclusions the answers to the three research questions they raised in the introduction.

As a final comment, I suggest to carefully consider if the conceptual framework and the case study should remain a part of the literature review or become a separate article. In fact, the description of the conceptual framework is very simple and does not allow the reader to understand properly what the author did and the results they obtained. A work fully dedicated to it would allow readers to properly appreciate the work done by the authors understanding all the necessary details.

---

## Referee Report (RR2)

**Minor comments**

List of affiliations: add the affiliation number

Line 4: including the methodologies for calculating natural hazards' indices

Line 81: check the citation format. I suppose it should be (Liberati at al., 2019)

Line 86: check the same citation as in line 81

Line 132: insert an empty line before "Food security", which is a new definition

Line 176: check the dot after 2011

Line 281: Li, Z., Zhang, Z., Zhang, J., Luo, Y., and Zhang, L. (2021). A new framework to quantify maize production risk from chilling injury in Northeast China. Climate Risk Management, 32. doi:10.1016/j.crm.2021.100299. This study proposes a Chilling Index to evaluate the effects of low temperatures on maize. However, it does not show the application of an index insurance scheme even if in the conclusion the index is considered promising for possible practical applications in the insurance field.

Line 300: basis risk

Line 314: check the brackets on Martinez Salguerio

Line 345 and following: Monteleone, B., Borzí, I., Bonaccorso, B., and Martina, M. (2022). Quantifying crop vulnerability to weather-related extreme events and climate change through vulnerability curves. Natural Hazards, (0123456789). doi:10.1007/s11069-022-05791-0 This study shows crop vulnerability functions for various weather-extremes and reviewed the methodologies used to derive them, among which crop modelling.

Line 357 to define

Line 366: remove 365

Line 374-375: It is well known that climate variables present a certain degree of uncertainty when they are predicted. Therefore, this aspect needs to be considered when estimating losses caused by climate-related events (Smith and Matthews, 2015)

Line 406: Figueiredo, R., Martina, M. L. V, Stephenson, D. B., and Youngman, B. D. (2018). A Probabilistic Paradigm for the Parametric Insurance of Natural Hazards. Risk Analysis, 38(11), 2400–2414. doi:10.1111/risa.13122. This study, even if not focused on crop insurance, provides a nice overview of basis risk and the issue you are addressing in the paragraph.

Line 427 and 429: check the brackets on Turvey et al. and Sacchelli et al.

Lines 425-430: please refer explicitly to figure 2 to enable the reader to clearly understand the meaning of the paths mentioned in the text.

Line 491: check the font of the citation (Bhering et al.)

Line 507: check citeppeterson2001report

Line 517: close the bracket after (values below K)

Line 525: model M1

Line 539: it seems from the following lines and Table 4 that you identified 6 clusters and not 5. Please correct or explain the reason for which you state that you identified 5 clusters.

Line 559: it is possible

Line 571: I do not find figure 7. Maybe you would like to cite Figure 5?

Line 614: testing should not have a capital letter

---

## Author Response (AR3)

Dr. Vassiliki Kotroni
Editor

*Natural Hazard and Earth System Sciences*

February 14th, 2022

Re: Review of manuscript egusphere-2022-498

Dear Dr. Vassiliki Kotroni,

We thank you for the diligent handling of our manuscript. We are very grateful for each comment and suggestion made by the referees, which improved our manuscript. We addressed all the points raised by referees in the table below:

| Observation | Our comment | Original text | Modifications made in the text |
|---|---|---|---|
| **Referee #1** | | | |
| L564: A further explanation is required about the sum of models. The method must be clear to the readers. Please explain that all 3 weather indices are examined as predictors of the losses associated with data from clusters 2,4,6….or something like that! | Thank you for the suggestion. We adapted and added further explanations about the sum of models. | The sum of the models (gray line in subfigure | Text: The sum of losses estimated with models M1 and M2 (Figure 5b) […]

Figure 5 caption: […] including the sum of the losses estimated with two loss models |
| L107: fifth step?? Inconsistent text compared to Figure 1 and previous text (L91). Overall many different numbering of steps is mentioned. E.g., again in L112… | Thank you for noticing the error. We rephrased the sentence | This evaluation excluded papers that did not provide information on index insurance design. Finally, in the fifth step, we critically reviewed the 26 most relevant studies.

The first crucial step for this analysis was defining the main concepts and definitions used to | The fourth step was performing a critical review of the 26 most cited papers published in the last five years of the dataset (2018 to 2022).

Before analyzing the full papers, it is critical to specify the main concepts and definitions |

| | | | |
|---|---|---|---|
| | | | analyze the material |
| L551-557 is very important but very difficult to follow. At least, the beginning should be corrected. The illustrative example does not suggest that….I assume it 'highlights' the following…Otherwise, this phrase is not connected well with the next, from a grammar point of view. | We thank you for the comment. We connected the text avoiding separating the arguments in numbering, and rewrote all the sentences. | The illustrative example suggests that… | The illustrative example highlights the multi-hazard nature effects of extreme weather events on crop yield losses. We used cluster analysis to identify what hazards were dominant each year that a crop loss event occurred [...] |
| Figure 2: state the exact period reviewed to show that 2022 was not fully covered | Thank you for the comment. We corrected it accordingly | | |
| Figure 5: Please correct TX090p. Also, the entire caption should be improved. The Latin numbering (i)(ii) is not shown in the Figure, and the other lines (observed, sum) are not included in the caption. | We rewrote the text in the legend to explain the models and include the other lines in the caption. | Risk analysis module applied to one specific location. Legend: (a): historic crop losses in the studied period for the city of Toledo; (b) crop loss probability in the studied period for the city of Toledo. (i) multi-hazard model M1(SPI,TX90p): using SPI and TX90p as inputs; and (ii) multi-hazard model M2(SPI,pmax,TX90p): using SPI, TX090p, and pmax as inputs | Risk analysis module applied to the city of Toledo for demonstrating the multi-hazard model M1(SPI,TX90p); multi-hazard model M2(SPI,pmax,TX90p), the sum of the two models; and observed crop losses in the studied period; (a) simulation of crop losses and (b) crop loss probability in the studied period including the sum of the losses estimated with two models |
| L77, correct: 'out' | Thank you for the comment. We corrected it accordingly | pointing ou limitations and recommended future works | pointing out limitations and recommended future works |
| Figure 1: Correct the legend of the 4th step | Thank you for the comment. We corrected it accordingly | | |

| | | | |
|---|---|---|---|
| L86: inconsistent writing ('by' is missing or parenthesis for the reference) | Thank you for the comment. We corrected it accordingly | We used a double-step analysis to analyze the data, following the PRISMA protocol Liberati et al. (2009) | We performed the literature review following the PRISMA protocol (Liberati et al. 2009) |
| L312: …interaction with what? | Thank you for the comment. We rewrote the sentence. | A multi-hazard approach requires understanding the frequency of each hazard and its interaction | A multi-hazard approach requires understanding the frequency and magnitude of multiple hazards and the possibility of them occurring simultaneously. |
| L354: both K & k are used for strike value? Please select between uppercase or lowercase | Thank you for the comment. We corrected it accordingly | critical variables (i) strike value K, and (ii) degree of coverage dc. The […] | critical variables (i) strike value K, and (ii) degree of coverage dc. The K […] |
| L374: correct 'were'. Is it 'when'? | Thank you for the comment. We corrected it accordingly | It is well known that climate variables present a certain degree of uncertainty were they are predicted | It is well known that climate variables present a certain degree of uncertainty when they are predicted |
| Figure 3. Correct 'communatary contract' in E2. I suppose the correct is 'community'? | Thank you for the comment. We corrected it accordingly | | |
| L536: correct the sentence | Thank you for the comment. We corrected it accordingly | The method applies a wavelet-based algorithm for multiple sites and requires and was applied using the R-package PRSim (Brunner et al., 2021)) | The stochastic simulation was performed using a wavelet-based algorithm that allows multi-site simulation. The simulation was implemented in R-Environment with the package PRSim (Brunner et al., 2021) |
| L527: correct 'clusters 2 adn 4' | Thank you for the comment. We corrected it accordingly | The multi-hazard model M1 was trained and validated using data from clusters 2 adn 4 | The multi-hazard model M1 was trained and validated using data from clusters 2 and 4 |

| | | | |
|---|---|---|---|
| L559: correct the sentence | Thank you for the comment. We corrected it accordingly | [...] severely impacted by extreme weather occurrences in Figure 5. t is [...] possible to observe that: | [...] severely impacted by extreme weather occurrences in Figure 5 it is [...] possible to observe that: |
| L567: please change the 'study case' to 'illustrative example' | Thank you for the comment. We corrected it accordingly | The study case in this subsection illustrates one possible application of the framework, | The illustrative example in this subsection illustrates one possible application of the framework, |
| L573-578: this paragraph is repeated…please delete | Thank you for the comment. We deleted the repeated paragraph | | |
| L588: Please rewrite. This sentence is not clear to me, including grammar errors: The vulnerability in the insurance design was characterized by selecting a loss models and defining threshold values that characterized loss events | Thank you for the comment. We corrected it accordingly | The vulnerability in the insurance design was characterized by selecting a loss models and defining threshold values that characterized loss events | The vulnerability analysis for insurance design was composed of the selection of a loss model selection and the definition of threshold values |
| L605: the sentence is not well-written. Please rephrase: Both excessive rainfall and high temperature and droughts and high temperatures were detected by the clustering analysis. | Thank you for the comment. We corrected it accordingly | Both excessive rainfall and high temperature and droughts and high temperatures were detected by the clustering analysis | Two examples of multi-hazard events were detected by the clustering analysis, one is the combination of excessive rainfall and high temperature, and the other is the combination of droughts and high temperature were detected by the clustering analysis |
| L606: put ': ' before 'the first was…', and correct for grammar | Thank you for the comment. We corrected it accordingly | The cluster model demonstrated that historic crop losses were divided into three groups, the | The cluster model demonstrated that historic crop losses were divided into three groups: the first was |

| | | first was precipitation deficit dominated, the second was precipitation deficit and high temperatures, and the third was excessive rainfall and high temperatures. | precipitation deficit dominated, the second was precipitation deficit and high temperatures, and the third was excessive rainfall and high temperatures |
|---|---|---|---|
| L610: Please rephrase. Grammar mistakes: … but also on analyzing the right data for right the hazard or multi-hazard selection | Thank you for the comment. We rewrote the paragraph correcting the grammar mistakes. | but also on analyzing the right data for right the hazard or multi-hazard selection | This example illustrates that the problem of the mismatch between actual losses and losses predicted from the index insurance contract, also called basis risk, does not depend only on having enough historical records of loss events but also on having enough understanding of what were the major drivers of loss events in the historical records |
| L614: correct 'Testing' with 'testing'. And 'specially' with 'especially' | Thank you for noticing the error. We corrected it accordingly | Testing different hypotheses of the interaction between hazards, specially | testing different hypotheses of the interaction between hazards, especially |
| L615: add 'and' before 'analyzing…' | Thank you for noticing the error. We corrected it accordingly | | and analyzing the trade-offs between loss model accuracy and the policyholders willingness to pay |
| Referee #2 | | | |
| List of affiliations: add the affiliation number | Thank you for the comment. We corrected it accordingly | | |
| Line 4: including the methodologies for calculating natural hazards' indices | Thank you for noticing the error. We corrected it accordingly | including a methodology for calculating natural hazards' indices | including the methodology for calculating natural hazards' indices |

| | | | |
|---|---|---|---|
| Line 81: check the citation format. I suppose it should be (Liberati at al., 2019) Line 86: check the same citation as in line 81 | We are sorry about this formatting error. We updated the citation. | | We performed the literature review following the PRISMA protocol (Liberati et al., 2009) |
| | | | |
| Line 132: insert an empty line before "Food security", which is a new definition | Thank you for the comment. We corrected it accordingly | | Corrected |
| Line 176: check the dot after 2011 | | According to Cobo et al. (2011). Density is a measure … | According to Cobo et al. (2011), density is a measure ... |
| Line 281: Li, Z., Zhang, Z., Zhang, J., Luo, Y., and Zhang, L. (2021). A new framework to quantify maize production risk from chilling injury in Northeast China. Climate Risk Management, 32.doi:10.1016/j.crm.2021.100299. This study proposes a Chilling Index to evaluate the effects of low temperatures on maize. However, it does not show the application of an index insurance scheme even if in the conclusion the index is considered promising for possible practical applications in the insurance field. | Thank you for the suggestion. This paper is very interesting and is adequate for studying temperature variation hazards. However, we will not include this paper in the discussion because we focused exclusively on indices for index-based insurance design. | | |
| Line 300: basis risk | Thank you for noticing the error. We corrected it accordingly | In order to lower the base risk | In order to lower the basis risk |
| Line 314: check the brackets on Martinez Salguerio | Thank you. We corrected the citation style | The assumption of independence was considered prior knowledge by (Martínez Salgueiro, 2019) and Guo et al. (2019). | The assumption of independence was considered prior knowledge by Martínez Salgueiro (2019) and Guo et al. (2019). |

| | | | |
|---|---|---|---|
| Line 345 and following: Monteleone, B., Borzí, I., Bonaccorso, B., and Martina, M. (2022). Quantifying crop vulnerability to weather-related extreme events and climate change through vulnerability curves. Natural Hazards, (0123456789). doi:10.1007/s11069-022-05791-0 This study shows crop vulnerability functions for various weather-extremes and reviewed the methodologies used to derive them, among which crop modelling. | We thank you for the comment. We included this paper in the discussion. | | Monteleone et al. (2022) reviewed methods used to model the functional relationship between a given extreme event and crop losses. They also highlight the need for studying crop vulnerability to other less studied climate-related hazards, such as extreme temperatures. |
| Line 357 to define | Thank you for the suggestion. We corrected this grammar mistake. | will be the key to defining the premium | will be the key to define the premium |
| Line 366: remove 365 | Thank you for noticing. We removed this number. | 365 tails, leading to underestimating pure risk premiums. | tails, leading to underestimating pure risk premiums. |
| Line 406: Figueiredo, R., Martina, M. L. V, Stephenson, D. B., and Youngman, B. D. (2018). A Probabilistic Paradigm for the Parametric Insurance of Natural Hazards. Risk Analysis, 38(11), 2400–2414. doi:10.1111/risa.13122. This study, even if not focused on crop insurance, provides a nice overview of basis risk and the issue you are addressing in the paragraph. | We thank you for the suggestion. We adapted the text accordingly | | Figueiredo et al. (2018) proposed a probabilistic framework for tackling uncertainties in trigger selection and damage modeling, improving basis risk quantification, evaluation, and communication [...] |
| Line 427 and 429: check the brackets on Turvey et al. and Sacchelli et al. | Thank you for noticing. We corrected this editing mistake. | (Turvey et al., 2019), for example, followed a path A1-C1-D2-E1-F1, i.e., drought insurance (A1) with a static threshold (C1), a loss model based on losses projected by an index (D2), for single | Turvey et al. (2019), for example, followed a path A1-C1-D2-E1-F1, i.e., drought insurance (A1) with a static threshold (C1), a loss model based on losses projected by an index (D2), for single policyholders |

| | | policyholders (E1) for many farmers in a region (F1). (Sacchelli et al., 2018) provides another example | (E1) for many farmers in a region (F1). Sacchelli et al. (2018) provides another example |
|---|---|---|---|
| Lines 425-430: please refer explicitly to figure 2 to enable the reader to clearly understand the meaning of the paths mentioned in the text. | Thank you for the suggestions. We updated the text with the adequate citation of figure 3. | This framework elicits paths | This framework (Figure 3) elicits paths |
| Line 491: check the font of the citation (Bhering et al.) | Thank you for the suggestion. We corrected it accordingly. | of medium-grained soils such as cambisols (BHERING et al., 2009) | of medium-grained soils such as cambisols (Bhering et al., 2009) |
| Line 507: check citeppeterson2001report | Thank you for noticing this editing mistake. We corrected it in the text accordingly | Climate Change Detection and Indices (ETCCDI) citeppeterson2001report | Climate Change Detection and Indices (ETCCDI) (Peterson et al., 2001) |
| Line 517: close the bracket after (values below K) | Thank you for noticing. We closed the bracket. | unfavorable crop yields for farmers ( (values below K and favorable crop yields (values above K) | unfavorable crop yields for farmers ( (values below K) and favorable crop yields (values above K) |
| Line 525: model M1 | We improved this sentence about model M1. Thank you for the suggestion. | Multi-hazard model 1 drought and thermal stressed using SPI and TX90p as inputs (M1(SPI,TX90p)); and (ii) Multi-hazard model excessive precipitation and thermal stressed model using SPI, TX90p, and pmax as inputs (M2(SPI,TX90p,pmax) | Multi-hazard model M1 drought and thermal stressed using SPI and TX90p as inputs (M1(SPI,TX90p)); and (ii) Multi-hazard model M2 excessive precipitation and thermal stressed model using SPI, TX90p, and pmax as inputs (M2(SPI,TX90p,pmax)) |
| Line 539: it seems from the following lines and Table 4 that you identified 6 clusters and not 5. Please correct or explain the reason for which you state that you identified 5 clusters. | Thank you for noticing it. We corrected the sentence. | We identified 5 clusters that are described in Table 4. | We identified 5 clusters that are described in Table 4. |

| | | | |
|---|---|---|---|
| Line 559: it is possible | We are sorry about this formatting error. We updated this part. | Figure 5. t is possible to observe that: | Figure 5 it is possible to observe tha |
| Line 571: I do not find figure 7. Maybe you would like to cite Figure 5? | Thank you for noticing the error, we wanted to cite figura 5 | | |
| Line 614: testing should not have a capital letter | We are sorry for the mistyping. Thank you for noticing it. | Testing different hypotheses | testing different hypotheses |